



# Emulating lateral gravity wave propagation in a global chemistry-climate model (EMAC v2.55.2) through horizontal flux redistribution

Roland Eichinger[1,2], Sebastian Rhode[3], Hella Garny[1,4], Peter Preusse[3], Petr Pisoft[2], Aleš Kuchař[5,*], Patrick Jöckel[1], Astrid Kerkweg[6], and Bastian Kern[1]

[1]Deutsches Zentrum für Luft- und Raumfahrt (DLR), Institut für Physik der Atmosphäre, Oberpfaffenhofen, Germany
[2]Charles University Prague, Faculty of Mathematics and Physics, Department of Atmospheric Physics, Prague, Czech Republic
[3]Forschungszentrum Jülich GmbH, Institut für Energie- und Klimaforschung, IEK-7, Jülich, Germany
[4]Ludwig-Maximilians-University Munich, Meteorological Institute, Munich, Germany
[5]Universität Leipzig, Institute for Meteorology, Leipzig, Germany
[6]Forschungszentrum Jülich GmbH, Institut für Energie- und Klimaforschung, IEK-8, Jülich, Germany
[*]now at: University of Natural Resources and Life Sciences (BOKU), Institute of Meteorology and Climatology, Vienna, Austria

**Correspondence:** R. Eichinger (roland.eichinger@dlr.de)

**Abstract.** The columnar approach of gravity wave (GW) parameterisations in weather and climate models has been identified as a potential reason for dynamical biases in middle atmospheric dynamics. For example, GW momentum flux (GWMF) discrepancies between models and observations at $60°$S arising through the lack of horizontal orographic GW propagation is suspected to cause deficiencies in representing the Antarctic polar vortex. However, due to the decomposition of the model domains onto different computing tasks for parallelisation, communication between horizontal grid boxes is computationally extremely expensive, making horizontal propagation of GWs unfeasible for global chemistry-climate simulations.

To overcome this issue, we here present a simplified solution approximating horizontal GW propagation through redistribution of the GWMF at one single altitude by means of tailor-made redistribution maps. To generate the global redistribution maps averaged for each grid box, we use a parameterisation describing orography as a set of mountain ridges with specified location, orientation and height combined with a ray-tracing model describing lateral propagation of so-generated mountain waves. In the global chemistry-climate model (CCM) EMAC (ECHAM MESSy Atmospheric Chemistry), these maps then allow us to redistribute the GW momentum flux horizontally at one level obtaining an affordable overhead of computing resources. The results of our simulations show GWMF and drag patterns which are horizontally more spread-out than with the purely columnar approach, GWs now also are present above the ocean and regions without mountains. In this paper, we provide a detailed description of how the redistribution maps are computed and how the GWMF redistribution is implemented in the CCM. Moreover, an analysis shows why 15 km is the ideal altitude for the redistribution. First results with the redistributed orographic GWMF provide clear evidence that the redistributed GW drag in the Southern Hemisphere has the potential to





modify and improve Antarctic polar vortex dynamics, thereby paving the way for enhanced credibility of CCM simulations
and projections of polar stratospheric ozone.

## 1  Introduction

The middle atmosphere is an important part of Earth's atmospheric system, housing the ozone layer and global scale dynamic
processes. This includes the Brewer-Dobson circulation (BDC, Brewer, 1949; Dobson, 1956), which controls the transport of
radiatively active trace gases in the stratosphere and mesosphere. Moreover, middle atmosphere dynamics are a notable source
of uncertainties for decadal to centennial climate projections as well as for medium-range weather forecasts due to dynamical
downward coupling mechanisms (see e.g. Hardiman and Haynes, 2008; Gerber et al., 2012; Hitchcock and Simpson, 2014).
In the high latitudes, these coupling mechanisms are strongly connected to strength and stability of the polar vortices. The
momentum and energy that dissipate when upward propagating atmospheric waves break are crucial for the polar vortices
and for other dynamical phenomena such as the semi-annual oscillation (SAO, Baldwin and Dunkerton, 2001) and the quasi-
biennial oscillation (QBO, Giorgetta et al., 2002) in the middle atmosphere (e.g. Charney and Drazin, 1961; Andrews et al.,
1987; Šácha et al., 2019). These atmospheric waves are commonly induced in the troposphere and propagate upward, thereby
transporting momentum and energy to the middle atmosphere. The global spectrum of atmospheric waves spans from small-
scale gravity waves (GWs) with wavelengths of the order of ∼10-1000 km to large-scale planetary waves. The resolution
of current climate models is fine enough to capture the planetary waves and their propagation and dissipation. GWs, on the
other hand, occur on small spatial scales and short time scales and at least parts of the GW spectrum cannot be resolved by
current state-of-the-art climate models and therefore have to be parameterised. GW parameterisations in general circulation
models (GCMs) are commonly subdivided into non-orographic GWs and orographic GWs (OGWs), referring to different
GW sources. The latter explicitly concerns mountain waves (MWs) originating from flow over orography, while the former
takes into account all other GW sources, e.g. convection, frontal instabilities or spontaneous adjustment (see e.g. Fritts and
Alexander, 2003; Alexander et al., 2010). Both GW schemes were historically included into weather and climate models and
tuned in order to eliminate or alleviate dynamical model biases (see e.g. Palmer et al., 1986; Giorgetta et al., 2002; Shepherd,
2007). OGWs for example helped to separate the stratospheric polar night jet from the tropospheric subtropical jet (Kim et al.,
2003; Alexander et al., 2010; Eichinger et al., 2020). Today, various versions of these parameterisation schemes (e.g. Lott and
Miller, 1997; Gregory et al., 1998; Scinocca and McFarlane, 2000) are still routinely applied in GCMs for climate simulations,
but as the understanding of GW processes grows, parameter specifications are constantly further developed (e.g. de la Cámara
et al., 2014; Garcia et al., 2017; Xie et al., 2020; Plougonven et al., 2020; van Niekerk and Vosper, 2021; Xie et al., 2021;
Ribstein et al., 2022).

The main interactions of GWs with prognostic quantities in GCMs is de- and acceleration of the winds at the location of
wave breaking as well as energy transfer in form of temperature and needs to be parameterised as correctly as possible. GWs
not only propagate upwards, but also horizontally and this lateral GW propagation has multiple effects (Song et al., 2007; Fritts
et al., 2016; Samtleben et al., 2019; Strube et al., 2021). First and foremost, it leads to a relocation of GWMF and therefore GW



drag. An indirect consequence is that GWs propagate upwards in other locations and thereby may encounter critical levels at different altitudes. Since GW drag is a function of GWMF and inverse density, which is decreasing with altitude, it potentially also leads to a de- or increase in total GW drag depending on the altitudes of GW breaking (e.g. Xu et al., 2017).

GW parameterisations in GCMs are implemented in a purely columnar manner, not allowing any horizontal propagation of GWs and their momentum. This is despite the fact that numerous studies have analysed lateral GW propagation (e.g. Preusse et al., 2002; Sato et al., 2012; Kalisch et al., 2014; Fritts et al., 2016; Perrett et al., 2021; Strube et al., 2021) and GWMF discrepancies between models and observations could largely be attributed to this process, particularly at $60°S$ (Geller et al., 2013; Kruse et al., 2022). Moreover, the spatio-temporal OGW distribution has been shown to be important for planetary-scale

wave fields via modification of refraction conditions (Šácha et al., 2016, 2021; Samtleben et al., 2019, 2020) and the lack of horizontal propagation has been suggested to be a source for zonal wind biases in the Antarctic polar vortex (McLandress et al., 2012; Gupta et al., 2021). However, due to the decomposition of the model domains onto different computing tasks for parallelisation, communication between horizontal grid boxes can become computationally extremely expensive, making lateral propagation unfeasible in multi-decadal global climate simulations. For example, Song et al. (2007) have implemented

a spectral parameterisation of GW drag induced by cumulus convection in a 3-dimensional framework in a Chemistry-Climate Model (CCM). Likely due to the computational cost of the development, this could not be established for operational use in the model.

To overcome this issue, while retaining computational efficiency and flexibility, we here present a solution that emulates lateral GW propagation through redistribution of the orographic GWMF in a CCM at a single altitude with tailor-made redis-

tribution functions. These global redistribution functions describe a general propagation pattern of MWs and are generated by use of a mountain wave model (MWM), which was developed particularly for this purpose and is described in detail by Rhode et al. (2023). There, the authors showed the capability of the model to reproduce lateral propagation patterns found in satellite and high-resolution simulations. For example, strong leeward OGW propagation has been found, which is in agreement with Sato et al. (2012). Within this MWM, an orography is described as a set of idealised mountain ridges, of which each excites

a specific MW. The idealised mountain shape allows estimation of the location, orientation, horizontal wavelength and displacement amplitude of launched MWs. The temporal and spatial propagation of these MWs is calculated using the ray-tracer GROGRAT (Gravity-wave Regional Or Global Ray Tracer; Marks and Eckermann, 1995; Eckermann and Marks, 1997). In the global CCM EMAC (ECHAM MESSy Atmospheric Chemistry, v2.55.2, Jöckel et al., 2010, 2016), these maps then allow redistribution of the GWMF horizontally, obtaining an affordable overhead of computing resources, independent of the model

decomposition in grid point space.

A schematic of our approach is shown in Fig. 1. First, the redistribution of GWMF is determined by means of a ray-tracing model. To do so, GWs are initialised from each source grid cell, and their locations (lat, lon) are detected and weighted by their GWMF at a fixed altitude, the target height $H_{\text{tar}}$. These locations are then taken as target location of the GWMF transport. In EMAC, this redistribution map ($\mu_{rd}$) is applied at a different height, the redistribution height $H_{\text{rd}}$, where the total

column GWMF is redistributed to the predetermined target locations with the given weight. This approximates the horizontal





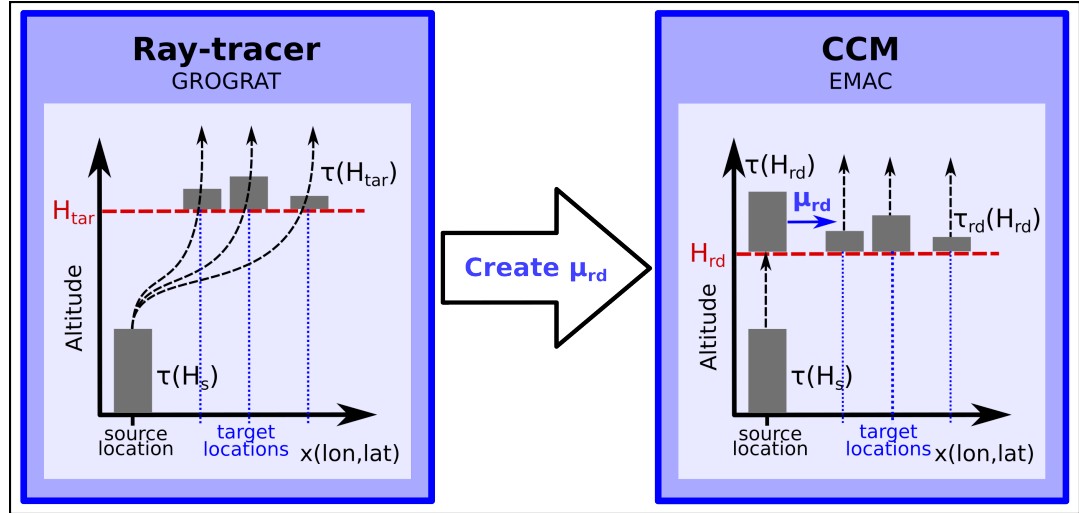

**Figure 1.** Schematic of the redistribution approach with example momentum propagation paths. The momentum transport target locations are estimated at a fixed target height, $H_{\mathrm{tar}}$ and weighted by the GWMF of the respective GWs. This pattern of horizontal transport (the redistribution map $\mu_{rd}$) is applied in EMAC at the redistribution height, $H_{\mathrm{rd}}$, which should be lower than $H_{\mathrm{tar}}$. Energy conservation is preserved by only redistributing the GWMF that EMAC computes in the source column just below $H_{\mathrm{rd}}$, i.e. by $\tau(H_{rd}) = \sum_x \tau_{rd}(H_{rd})$.

momentum transport of the ray-traced GWs from lateral propagation. The redistribution height should be lower than the target height for a better approximation of the GW propagation pattern in the CCM.

In Sect. 2, we describe the employed ridge parameterisation and the coupled ray-tracer GROGRAT. Further, the generation of the redistribution maps with this model system is presented and exemplary redistribution patterns are shown. Moreover, analyses to determine the ideal ray target height for the generation of optimal redistribution functions and the ideal altitude at which the redistribution should be applied in the CCM are given. In Sect. 3, implementation of usage of the redistribution functions to horizontally redistribute orographic GWMF in the CCM EMAC is described and the details in the used Lott and Miller (1997) orographic GW parameterisation scheme are presented. Moreover, a scaling analysis to evaluate the additionally required computing time and an update to the subgrid-scale orography to high-resolution elevation data is provided. The section proceeds with the results of test simulations with the new CCM implementations and with a brief analysis of their influence on Antarctic polar vortex dynamics. To wrap-up the paper, a summary and several discussion points are presented in Sect. 4.

## 2 Generating redistribution maps with a Mountain Wave Model

Here, redistribution maps that describe an approximated propagation pattern of OGWs are here generated by means of a Mountain Wave Model (MWM). This is done in three steps: (i) mountain ridge identification for MW source estimation, (ii) ray-tracing for propagation of MWs through the atmosphere and (iii) evaluation of the results to generate redistribution maps. These three steps are described in detail in the following sections. The idea is that the redistribution maps are linear



transformations of GWMF fractions from source grid cells to target grid cells. The values of the transformation are proportional to the amount of GWMF that has been transported from a source to the corresponding target grid cell. Additionally, in Sect. 2.4 an assessment of ideal values for free parameters in these transformations is presented.

## 2.1 The ridge parameterisation

First, we apply an algorithm to detect and parameterise mountain ridges from topographic data to retrieve MW parameters and to analyse the MW spectrum and distribution that needs to be initialised in the ray-tracing model. The approach applied here is described in detail in the companion paper Rhode et al. (2023) and it is similar to the one performed by Bacmeister (1993) and Bacmeister et al. (1994). The used topography data set 'ETOPO1' (Amante and Eakins, 2009) features a 1 arcmin resolution.

The general idea is to represent the topography in terms of a small number of idealised Gaussian-shaped mountain ridges. For this, different bandpass filters between about $100\,\mathrm{km}$ and $1500\,\mathrm{km}$ and of varying width are applied to the topography for scale separation. A skeleton of the bandpass filtered topography is generated by a reduction to the ridge lines of the field, represented by the mountain crests. Locations where idealised ridges will be fitted are singled out using a Hough Transformation. This results in approximately straight mountain ridge location candidates. At these, a fit with a Gaussian-shaped mountain ridge (Gaussian shape across the mountain, constant along the ridge) is performed in order to minimise the absolute difference of the idealised ridge to the bandpass filtered topography. Applying this algorithm to all spectral bands results in a ridge collection of Gaussian-shaped mountains that approximate the underlying topography in the chosen length-scales.

Further, it is assumed that each of these mountains excites a MW with displacement amplitude $\delta$ proportional to the best-fit height $h$ (for sufficiently strong winds $\delta = \frac{h}{2}$) and horizontal wavelength $\lambda_{\mathrm{hor}} = 2\pi\sigma$, where $\sigma$ denotes the width of the Gaussian. A possible reduction in amplitude due to weak low-level winds, i.e. wind blocking, is accounted for as well as a reduction in amplitude, if the flow is perpendicular to the ridge (e.g. Barry, 2008). For initialisation, MWs are assumed to always launch perpendicular to the source ridge. For more details and a validation of this approach, see Rhode et al. (2023).

## 2.2 The ray-tracer GROGRAT

For the propagation of initialised MWs, the ray-tracer GROGRAT (Gravity-wave Regional Or Global Ray Tracer; Marks and Eckermann, 1995; Eckermann and Marks, 1997) is used. GROGRAT is a global ray-tracing model describing the propagation, amplitude evolution and dissipation of atmospheric GWs within the limits of linear GW theory. It calculates 4-dimensional GW ray trajectories in a non-hydrostatic atmosphere by use of non-linear differential equations derived from the dispersion relation based on WKB (Wentzel-Kramer-Brillouin, e.g. Bretherton, 1966) theory. Under consideration of meteorological background fields, GROGRAT integrates these differential equations numerically to calculate the GW trajectories. The (horizontal wind) amplitudes are calculated from the vertical flux of wave action density under the assumption that this is a conserved quantity along the ray path except for saturation and breaking. Wave breaking is assumed if the wave amplitude is large enough to break local dynamic stability, i.e. the wave reaches its saturation amplitude, which is described by Fritts and Rastogi (1985). In addition, waves are damped along their trajectory due to turbulence and radiation.



In the present study, the trajectory calculations for the MWs initialised from the ridge collection of Sect. 2.1 are performed
using ERA5 reanalysis data (Hersbach et al., 2020) as background meteorological conditions. These data are provided on a
0.3° horizontal, 1 km vertical grid and with 6 h time resolution.

## 2.3 Redistribution maps

We apply the Mountain Wave Model (MWM), i.e. the ridge parameterisation coupled to the ray-tracer GROGRAT, as described
in Sects. 2.1 and 2.2, to generate global horizontal redistribution maps for use in the CCM EMAC in the next step. The ray-
tracer provides all necessary information to quantify momentum transport due to propagating MWs, i.e. location, amount of
GWMF and horizontal as well as vertical wavelengths and is therefore a suitable tool to generate the redistribution matrices
approximating the GWMF transport.

The redistribution of GWMF corresponds to a mapping from source grid cells to target grid cells with the transformation
values proportional to the amount of GWMF that has been transported from a source to the corresponding target grid cell.
Hence, we generate a 4-dimensional map $\mu_{\mathrm{rd}}(\phi_{\mathrm{src}}, \varphi_{\mathrm{src}}, \phi_{\mathrm{tar}}, \varphi_{\mathrm{tar}})$ describing the general MW propagation pattern. Here, $\phi$
and $\varphi$ denote latitude and longitude, respectively, the subscript src refers to the source grid cell, i.e. the location at which the
MW was excited, and tar refers to the target grid cell, i.e. the grid cell in which the MW's GWMF is transported to. The map
entries are generated by adding up the GWMF in the corresponding target location at a given altitude $H_{\mathrm{tar}}$ from all source
cells for each MW. Since the ray tracer provides only the peak value of GWMF and we need to account for the total GWMF
of the wave, we scale the contribution of each MW by its horizontal wavelength and the length of the source ridge exciting
the particular wave. This considers the effect that waves with a wider horizontal extent have a more spread-out footprint in
physical space than small localised ones of the same peak GWMF. The latter is the case if we assume that GW packets extent
the same number of wavelengths at any scale, which is reasonable because we only consider GWs of a single origin (mountain
waves). The resulting redistribution map is scaled such that

$$\sum_{\phi_{\mathrm{tar}}, \varphi_{\mathrm{tar}}} \mu_{\mathrm{rd}}(\phi_{\mathrm{src}}, \varphi_{\mathrm{src}}, \phi_{\mathrm{tar}}, \varphi_{\mathrm{tar}}) = 1 \ \ \forall \ \ (\phi_{\mathrm{src}}, \varphi_{\mathrm{src}}). \tag{1}$$


This normalization ensures energy conservation in the sense that we only redistribute what the model already initialised.
Thus, for each source grid cell, we end up with a probability map that defines to which target cells which fraction of the
GWMF is transported to. For consistency this normalisation is enforced also for source cells where no MWs are initialised
(e.g. above the ocean) by not assuming any horizontal transport, i.e. $\mu_{\mathrm{rd}} = 1$ at the source grid cell and 0 in all other grid cells.
The effect of the target height $H_{\mathrm{tar}}$, which is the height at which the position of the MW is taken as transport target, on the
redistribution maps and resulting redistribution maps is investigated in Sect. 2.4.

To generate the redistribution maps, global fields of the zonal and meridional wind components, temperature and pressure
are required as meteorological background conditions. For this, data from models or reanalyses can be taken, here we chose the
ERA5 reanalysis (Hersbach et al., 2020). As an example for the redistribution (or probability) maps, Fig. 2a shows the target
values of the redistribution for MWs initialised at 48.8°S, 70.3°W. This example redistribution map was produced from one
month of ERA5 data for July 2006 with a target altitude of 40 km. The redistribution map shows that only little of the GWMF



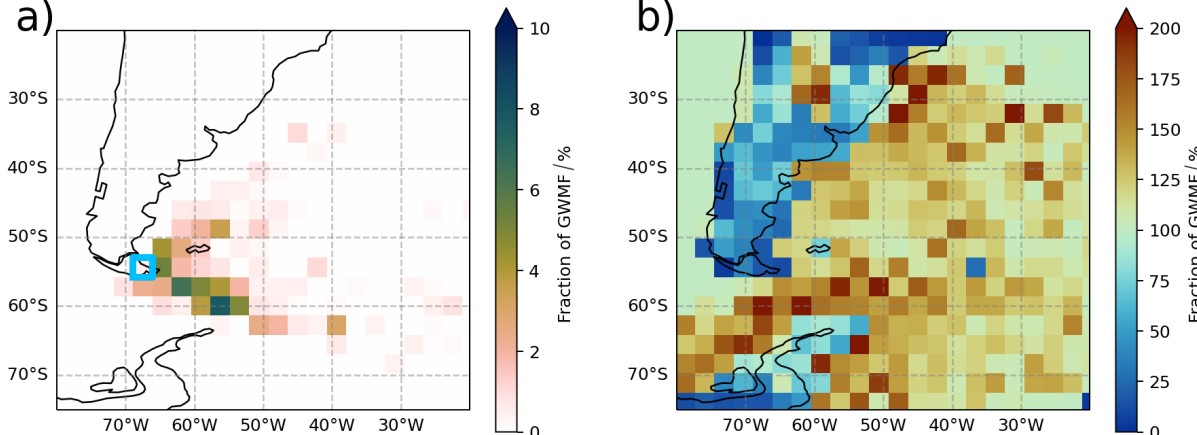

**Figure 2.** a) Exemplary redistribution function for the source grid box at $48.8°S$, $70.3°W$ (light blue square). Colour shading shows the target cell percentage of the GWMF transported from this source grid. b) Redistribution map summed over all source grid boxes to provide an estimate of the total GWMF redistribution if all source grid cells would exert the same amount of GWMF. The colour shading corresponds to the percentage of GWMF ending up in the respective grid cell, blue cells result in reduced and red cells in increased GWMF values.

that originates from the source grid cell will stay in this box, and therefore in the model column, after redistribution. Most of it will be redistributed to grid cells downwind, a large part of which to the South-East towards the Drake Passage. Therefore, the CCM should show increased drag at $60°S$ after redistribution. The rather large spread of GWMF transport can be attributed to
a variety of different wind situations. But also the relatively large grid size used here ($\sim 2.8° \times 2.8°$) leads to inclusion of many differently oriented ridges in one cell. This grid size was chosen as it represents the standard EMAC horizontal resolution setup of T42. Moreover, Fig. 2b shows the same redistribution map summed over all source grid cells to display the total redistribution from all grid boxes. It is an indicator of where to expect enhanced and reduced GWMF (considering only relative in- and outflow). As expected, we see a reduction above land as well as an enhancement above the ocean and the transport is
mainly in the direction of the predominantly eastward winds. But in the northern parts of the shown section, there is also a small region above land with values larger than 100%. As the percentage values of the cumulative redistribution fractions of all source grid cells are depicted in Fig. 2b, this, however, does not necessarily mean an increase of GWMF after redistribution.

### 2.4 Assessment of redistribution maps and ideal height parameters

As indicated above and illustrated in Fig. 1, the implementation of the GW redistribution in EMAC depends on the two
parameters target height, $H_{\mathrm{tar}}$, and redistribution height, $H_{\mathrm{rd}}$. The target height defines the altitude at which the horizontal target location is allocated to generate the redistribution maps with the ray-tracer after GWMF propagation. In general, the higher this altitude is chosen, the further the GWs will have propagated from their source. This can be used to tune the redistribution map w.r.t. the locations and altitudes of GW activity. The redistribution height defines the level at which the redistribution of GWMF is performed in EMAC. In other words, it is the altitude above which GWMF in EMAC is not





restricted to the source location any longer, but instead distributed to the locations determined by the redistribution map. And these locations again, are estimated from the MWM at the target height.

To find the ideal estimates for these parameters, we approximate the effect of GW redistribution on GWMF distributions using the MWM. For this, we firstly apply the MWM in a columnar manner by restricting the GWs to their source location, i.e. without lateral propagation, in the following called 'NO_HOR'. Secondly, we apply the GW redistribution in the columnar-

mode MWM in 9 simulations with varying $H_{\mathrm{tar}}$ (10-50 km in 5 km steps), mimicking the implementation in EMAC that will be described later in Sect. 3. Thirdly, we use the MWM in the configuration it was used to generate the redistribution maps, allowing for full lateral propagation of GWs. The latter will be considered as our ground truth for this assessment, i.e. used as reference and called 'REF'.

Fig. 3 shows vertical profiles of GWMF and GW drag deviations of the NO_HOR simulation and the simulations with GW

redistribution w.r.t. the fully propagating REF simulation. The deviations are taken as the relative deviation to REF at each grid point location and averaged globally and over the month.

First, we look at the results from the simulations of July 2006 applied to the same month of data (Fig. 3a-c). Figure 3a shows improved agreement in GWMF (always with regard to REF) through GW redistribution. In the upper stratosphere, the mean deviation in GWMF drops from 140% in the columnar case to about 60% depending on the target height. Similarly for the drag

in Fig. 3b, the deviation drops from about 160% to 80-90%. There is a trade-off evident here: the higher the target height, the better the agreement of the wave field in the upper atmosphere, but the more overestimated the propagation of MWs at lower altitudes. Since dissipating MWs primarily interact with the general circulation by exerting drag and thus changing the model winds, our main objective is to achieve as correct as possible the relocation of the (total) drag.

As atmospheric density is decreasing with altitude, the relative deviations as shown in Fig. 3b will have a stronger effect in

the upper atmosphere. Therefore, Fig. 3c shows the reduction in total deviation with respect to the columnar simulation, which is proportional to the area between the black and the coloured curves of Fig. 3a and b (scaled with the respective GWMF/drag at each altitude level). The reduction in deviation by redistribution is equivalent to a reduction in area between the GWMF/drag profile and zero. The best representation of GWMF transport is achieved with a target height of about 25 km. This rather low altitude is a consequence of GWMF of primary MWs diminishing with altitude in general. In contrast to this, the optimum for

drag is reached at a target height of about 40 km. As most MWs dissipate their momentum in the upper stratosphere, our main focus here is improving GW drag in these altitudes.

Next, we repeat the assessment with a redistribution map that was generated from propagation patterns averaged over year 2006 to yield a more general result. As before, the redistribution maps were applied to simulations of July 2006. The results of these simulations (Fig. 3d and e for GWMF and drag, respectively) are similar to the redistribution map generated from

July data. After GW redistribution, the GWMF deviations are larger than with the columnar approach below around 15 km, but smaller above. The same applies to GW drag, where the deviation is reduced particularly around 10-20 km. This is where the summertime stratospheric wind reversal is located, hence possibly causing an impact in model dynamics through GW redistribution.





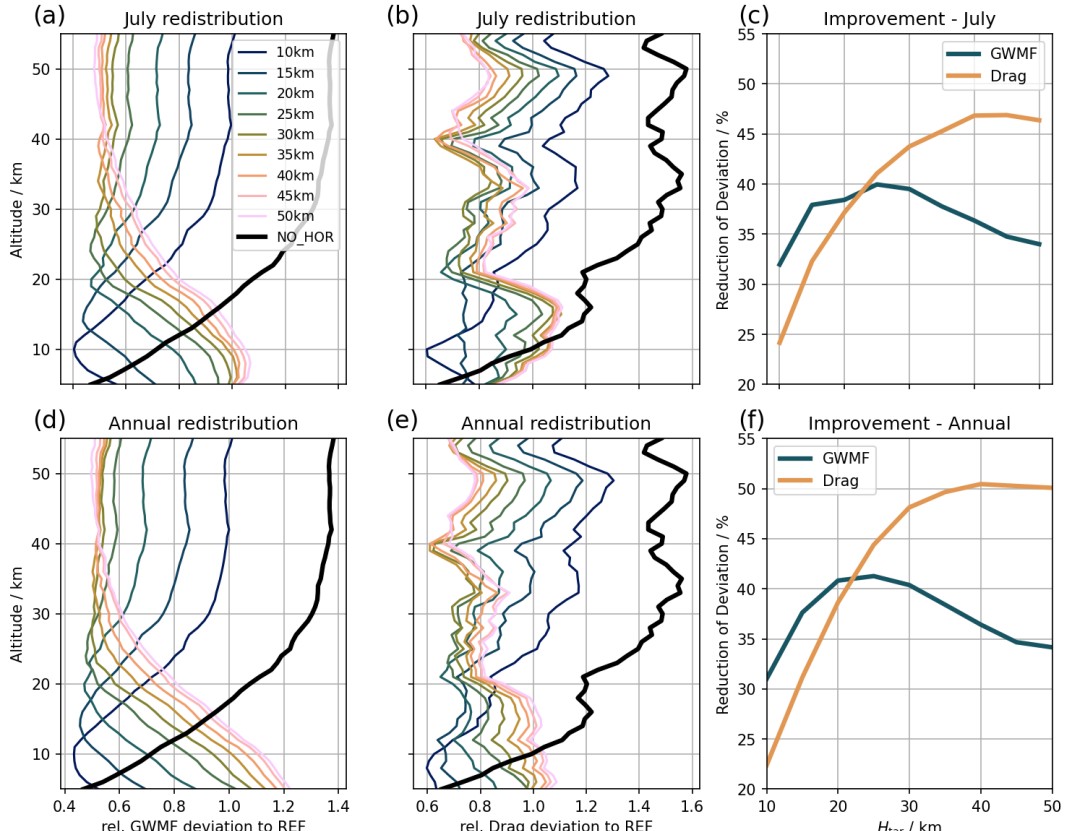

**Figure 3.** Comparison of MWM simulations with GW redistribution (with varying $H_{\mathrm{tar}}$, coloured lines) and vertical-only propagating MWM simulations (NO_HOR, black lines) to a simulation including horizontal propagation (REF) for a) GWMF and b) GW drag. The horizontal axis shows the relative deviation to REF at the corresponding altitude in monthly and global mean. Panel c shows the maximum improvement (reduction of deviation to REF) one can expect for different $H_{\mathrm{tar}}$ with optimal redistribution height. This is proportional to the area between the black and the coloured curves in panels a and b. The redistribution map used here was generated from data of July 2006 and applied to the same month for panels a-c and for a redistribution map calculated for the full year 2006 in panels d-f.

For the cumulative improvement across all altitudes using the annual mean redistribution map, Fig. 3f shows the reduction in total deviation. The optimal target height is still 25 km considering GWMF and 40 km for GW drag, whereas we consider GW drag substantial for our purpose. Therefore, we determine the target height to be 40 km in all following simulations. Note that the total agreement in GWMF and GW drag transport is better with the annual redistribution map compared to the July-only redistribution map. This can be explained by improvements in the summer hemisphere as well as by an increase in statistics, as there are now twelve times as many ray-traces considered.

The next question is where the ideal redistribution height is for a fixed $H_{\mathrm{tar}}$ at 40 km. For this, we search the minimum difference to the REF simulation through GW redistribution, i.e. the altitude where the coloured lines in Figs. 3b and e cross



the black line. For the considerations in Fig. 3, the redistribution height was chosen case specific at the cross-over point, i.e. the height, above which a redistribution yields a net positive improvement. Physically speaking, it might be advantageous to implement such a dynamic redistribution height. However, to keep complexity and computing time low, we aim at a fixed redistribution height that leads to the highest average improvement throughout the year and therefore we will look at the parameter sensitivity in the following.

Fig. 4a shows the monthly average improvement in GW drag through GW redistribution in dependence of month and redistribution height. Here, the more general annually averaged redistribution map has been used. The largest improvements are achieved during austral winter, which could be expected due to the strong lateral propagation patterns in the Southern Hemisphere. These improvements are not very sensitive to the redistribution height, only from January to March the approximation benefits substantially from a lower $H_{\mathrm{rd}}$. In general, the improvements are above 33% throughout the year (with a minimum in February) if $H_{\mathrm{rd}}$ is chosen at or below 15 km.

The annual mean improvement in GWMF and GW drag through GW redistribution is shown in Fig. 4b. For GW drag, we see an almost constant improvement of around 42% until a $H_{\mathrm{rd}}$ of about 15 km, followed by a slight decrease of improvement for higher $H_{\mathrm{rd}}$ altitudes, which is due to the growing underestimation of GW redistribution. The annual mean improvement in GWMF shows a much stronger dependence on $H_{\mathrm{rd}}$. The improvement strongly increases until a $H_{\mathrm{rd}}$ of ∼15 km and decreases again with larger $H_{\mathrm{rd}}$. Since GWMF is constant with altitude until wave breaking happens (and not scaled inversely with density as GW drag), GWMF is more sensitive to over- and underestimation of horizontal propagation at lower altitudes. To obtain the best improvement for GW drag throughout the year, while still achieving a good approximation of lateral propagation, we therefore set the redistribution height to 15 km in all following considerations. Physically speaking, 15 km is an excellent result for $H_{\mathrm{rd}}$, because this is above the tropopause and below the (summer) wind reversal, which both are atmospheric regions with strong effects on GW attenuation.

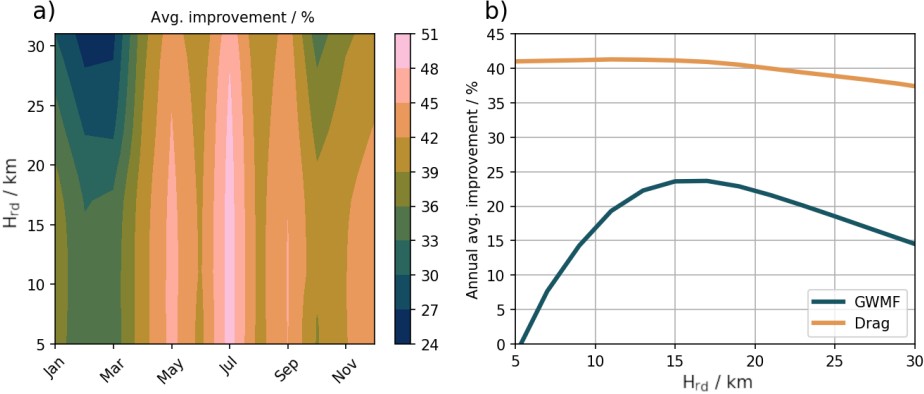

**Figure 4.** a) Improvement of GW drag across year 2006 as monthly means for varying $H_{\mathrm{rd}}$. The improvement is the reduction in deviation from REF through GW redistribution. b) Annual mean improvement for GWMF (blue) and GW drag (orange) with varying $H_{\mathrm{rd}}$.



## 3 Redistributing GW fluxes in the global CCM EMAC

In the present study, we use the redistribution maps described in Sect. 2 in the ECHAM/MESSy (European Center HAmburg Model / Modular Earth Submodel System) Atmospheric Chemistry (EMAC Jöckel et al., 2005, 2010, 2016) model. ECHAM (ECMWF Hamburg) is a global atmospheric GCM describing dynamics and comprising parameterisations for phyiscal processes (Roeckner et al., 2003, 2006). MESSy is a framework for standardised implementation of Earth System Models with flexible complexity, providing an infrastructure with generalised interfaces for coupling ESM components. The MESSy community policy and coding standard enables all users to optionally use all new developments, including those described in the present paper. Moreover, the flexible structure of MESSy allows to transfer the developments to base models other than ECHAM. The standard model resolution for our purposes is T42L90MA, which corresponds to a resolution of $\sim 2.8° \times 2.8°$ in latitude and longitude of the corresponding quadratic Gaussian grid and 90 layers in the vertical. In this setup, the uppermost level is located around 0.01 hPa and middle atmospheric dynamics (MA) are explicitly resolved. In all simulations presented here, only the basic MESSy submodels for dynamics, clouds and diagnostics are applied. In the CCM EMAC, the submodel "OROGW" comprises the subgrid-scale orography (SSO) scheme, including the OGW parameterisation. OROGW is the central part for the implementations of the GWMF redistribution described next.

### 3.1 Implementation

As stated above, lateral propagation of GWs cannot be realised computing cost-efficiently in state-of-the-art GCMs due to the communication overhead between horizontal grid boxes. The separation and parallelisation of computation of different model domains on different computing tasks results in high cost and time for communication between (adjacent) grid cells. Hence, the idea is to apply the GW redistribution only at one single altitude level, which shall account for the entire horizontal propagation of all GWs. Although this is a crude approximation, it is an important step towards a better representation of GW drag in the middle atmosphere. As described in Sect. 2.4, the ideal altitude for the GW redistribution is approximately 15 km, i.e. around 120 hPa. In the here used L90MA setup, the closest (hybrid sigma-pressure) model level to this altitude is level 65. This means, when the GW redistribution in EMAC is active, the levels 90 (bottom) to 66 still use the ordinary columnar OGW scheme, GW redistribution happens in level 65, and in the levels 64 to 1 (top), the redistributed GW fluxes again are treated by the columnar approach.

The submodel OROGW in EMAC comprises the OGW parameterisation by Lott and Miller (1997) and Lott (1999), which is described by Roeckner et al. (2003) for its use in ECHAM. In this scheme, GWMF $\tau$ (i.e. the Reynolds stress) in the level above the low-level breaking layer is launched by

$$\tau = \rho_H G U_H N_H Z_{\text{eff}}^2 \frac{\sigma}{4\nu} |\overrightarrow{P}|, \tag{2}$$

where $\rho_H$, $U_H$ and $N_H$ are the incident density, zonal wind component and Brunt-Väisälä frequency at the launching height $H$, respectively, $G$ is the GW parameter, which tunes the GWMF and has its default value set to 0.8, $\nu$ and $\sigma$ are the standard deviation and the slope of the SSO, respectively, $Z_{\text{eff}}$ is the effective mountain height determined by the mountain height above



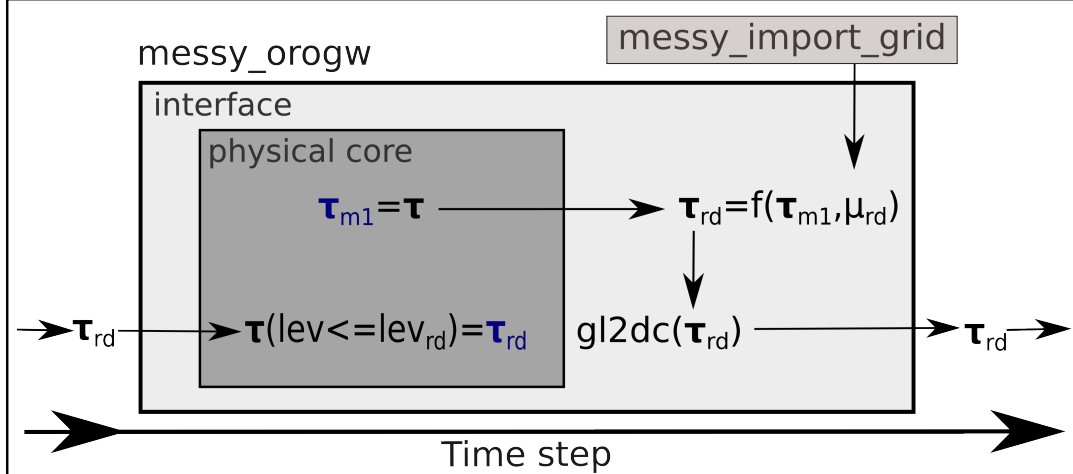

**Figure 5.** Schematic of the GW redistribution implementation in the EMAC submodel OROGW. The physical core denotes the parameterisation itself and the interface is the connection to the other model parts. gl2dc stands for global to decomposition, $\mu_{rd}$ for the redistribution (rd) function, m1 for the minus 1 (w.r.t. the time step) value, lev for model level and $\tau$ for the GW flux. See main text for more explanations and the Supplement for EMAC/MESSy-specific implementation details.

the blocked flow and $|\overrightarrow{P}|$ is the directional vector determining the angle between the incident flow and the normal orographic ridge direction. This GWMF is calculated from the bottom upwards until either a critical level is encountered, leading to full deposition of GWMF, or partial GW breaking due to saturation happens.

The below described implementation of the OGW flux redistribution is schematically depicted in Fig. 5. Some EMAC/MESSy-specific implementation details are further provided in the Supplement.

In order to perform the redistribution, first $\tau$ at the redistribution level (here 65) is transferred from the physical core to the interface, where it can be processed further. This $\tau$ will be the basis for the redistributed flux. As this will happen in the subsequent time step, it is called $\tau_{m1}$, where m1 stands for time step minus 1. At the end of the time step, global fields are

generated by looping over the processing elements that are determined by the decomposition. This allows redistribution of $\tau_{m1}$ globally to yield the redistributed $\tau_{rd}$ by

$$\tau_{rd}(\phi,\varphi) = \sum_{\phi_{src},\varphi_{src}} \tau_{m1}(\phi_{src},\varphi_{src}) \cdot \mu_{rd}(\phi_{tar},\varphi_{tar},\phi_{src},\varphi_{src}), \tag{3}$$

where $\mu_{rd}(\phi_{tar},\varphi_{tar},\phi_{src},\varphi_{src})$ is the redistribution map that was described in Sect. 2.3 and the subscripts $tar$ and $src$ denote target and source of the GWs. The 4-dimensional redistribution map (plus a possible time dimension) is read-in via

IMPORT_GRID (Kerkweg and Jöckel, 2015), which has been extended for the application of 4-dimensional arrays (plus time axis) for this purpose. Next, a function (gl2dc) is applied to transpose $\tau_{rd}$ from a global field to a field in the simulation-specific decomposition. This function comprises the major part of the redistribution-caused overhead relating to total required comput-





ing time and memory loading on individual compute tasks, which will be further investigated in Sect. 3.3. $\tau_{rd}$ is then used in the subsequent time step to override $\tau$ in the level of redistribution and above before computation of wave breaking.


Wave breaking occurs when the critical Richardson number of 0.25 is reached. In this case, GWs are assumed to saturate, meaning that their amplitude is reduced to the value at which instability occurs (Lindzen, 1981). For this, in the parameterisation the saturated flux is calculated by

$$\tau_s = \frac{GU^2}{N^2} Z_{\mathrm{oro}} \qquad \text{with} \qquad Z_{\mathrm{oro}} = \rho_H N_H U_H Z_{\mathrm{eff}}^2 \frac{\sigma}{2\nu}. \tag{4}$$

Here, $Z_{\mathrm{oro}}$ contains parameters describing the wave properties at launch level. In the columnar approach, these values always refer to the same grid cell. However, when applying the GW redistribution, the launch level properties have to be communicated to the new location the GWMF was distributed to. Otherwise, for example in grid boxes above the ocean, $Z_{\mathrm{oro}}$ would be zero and thus lead to erroneous complete breaking of the waves. This means, for correct calculation of GW breaking after GW redistribution, $Z_{\mathrm{oro}}$ has to be redistributed as well. However, in contrast to $\tau$, $Z_{\mathrm{oro}}$ is a (dynamic) parameter and not a variable

that can be added up. Therefore, $Z_{\mathrm{oro}}$ has to be normalised by the sum of all contributing entries of the redistribution map to account for the fraction of the parameter from a particular source box. Hence, redistributed $Z_{\mathrm{oro}}$ is calculated by

$$Z_{\mathrm{oro-rd}}(\phi_{tar}, \varphi_{tar}) = \frac{\sum_{\phi_{src},\varphi_{src}} Z_{\mathrm{oro-m1}}(\phi_{src}, \varphi_{src}) \cdot \mu_{\mathrm{rd}}(\phi_{tar}, \varphi_{tar}, \phi_{src}, \varphi_{src}),}{\sum_{\phi_{src},\varphi_{src}} \mu_{\mathrm{rd}}(\phi_{tar}, \varphi_{tar}, \phi_{src}, \varphi_{src})} \tag{5}$$

As GW drag is computed by the GW flux difference between the level in question and the level below, loss and gain of GW flux through redistribution would lead to erroneous GW drag. Therefore, wave breaking and computation of GW drag needed

to be disabled at the level of redistribution. Consequently, no GW drag can appear at this model level. This technically needed simplification needs to be kept in mind when analysing the results. Another option could be to fill this level with the drag of the level below or above. However, the effect of OGW drag at the redistribution altitude ($\sim$15 km) is generally relatively low in comparison to other effects. Therefore, the level without OGW drag does not generate any inconsistencies in the dynamical fields in our simulations, which gives us confidence that the chosen approach is applicable.

## 3.2 Updating the subgrid-scale orography

During conduction of the first test simulations with the GW redistribution, we identified deficiencies in the SSO representation in EMAC. In particular, we found the continental edges to be underrepresented in the standard SSO. This means that in comparison with the ridge parameterisation in GROGRAT, GW launching does not happen in grid boxes at the continental edges in EMAC. The MWM shows strong lateral propagation especially by MWs from these regions.

In order to achieve a better representation of MW launching in EMAC, we therefore updated the SSO by means of the same ETOPO1 (Amante and Eakins, 2009) topography data used for the ridge parameterisation in Sect. 2.2. While the original SSO data used in EMAC is based on a US Navy (10' × 10') data set (see Wallace et al., 1983), the new parameters are derived from the high-resolution 1' × 1' ETOPO1 data. Mean elevation, standard deviation, peaks (maximum) and valleys (minimum) were derived directly from the topography data within the corresponding model grid cells. The other three required parameters for





the parameterisation, namely anisotropy, slope and main orographic angle, can be derived from the former parameters based on the equations described by Lott and Miller (1997) and Baines and Palmer (1990). These are briefly repeated in the Supplement. Exemplarily, Fig. 6 shows the mean orography of the standard SSO and of the updated version for southern South America and the Antarctic Pensinsula. For completeness, all SSO parameters are shown for this region in the Supplement (Figs. S1 and S2).

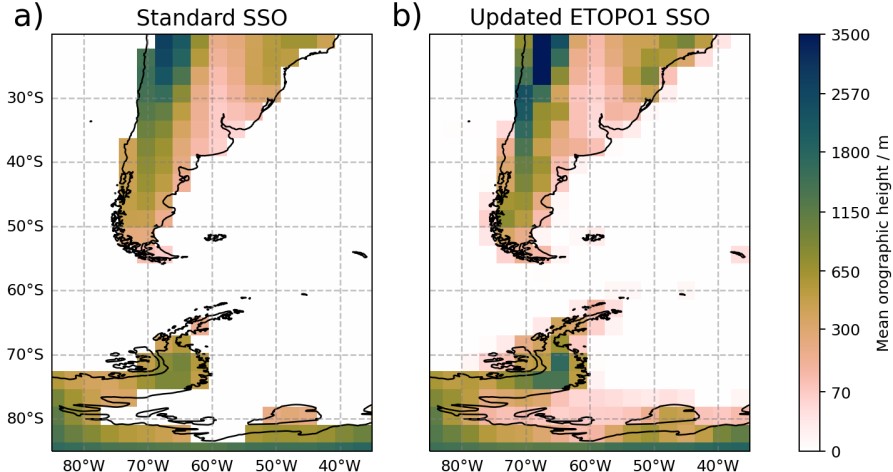

**Figure 6.** Mean subgrid-scale orography elevation as used in a) standard EMAC simulations and b) as updated from ETOPO1 high-resolution topography data. For details see main text.

Figure 6 shows that the updated SSO is more nuanced than the standard one and in particular, that the representation

of orography is enhanced at the continental edges. Especially the Antarctic Peninsula displays several grid boxes where no orography can be seen in the standard SSO, but the ETOPO1 SSO does show orographic elevation. Also some of the smaller islands, e.g. South Georgia, displays SSO in the updated version, which might contribute to a more realistic description of GW drag in the polar vortex (see e.g. Perrett et al., 2021). In the model test simulations which will follow in Sect. 3.4, we will investigate how this SSO update affects GW flux and drag in the middle atmosphere.

### 3.3 Run time performance analysis

To assess the additional computing resources required to apply the GW redistribution in EMAC, we performed 10 dedicated simulations over one simulation month. Five of these simulations were carried out with the GW redistribution activated (Red.), the other 5 without (Col.). The respective 5 simulations differ by usage of the number of compute tasks, i.e. 64, 128, 256, 512, 1024 tasks are used, respectively. See the Supplement (Tab. 1) for specific usage of the associated decomposition in these

simulations. We performed the simulations on HLRE-4 (levante) at the German Supercomputing Climate Center (DKRZ). The standard nodes of this high performance computer feature 128 tasks per node, hence our simulations used 0.5, 1, 2, 4 and 8 compute nodes, respectively. In these simulations, only the very basic setup for model dynamics and physics is activated. Moreover, all model output, except for the run time analysis output (QTIMER), which only outputs once at the end of the



month, is switched off to assess the relative effect of only the model performance itself without any additional output. Fig. 7
shows run time and required node-hours (node-h) of the ten simulations. Note that due to the output design, the first time step, which includes the build-up of the model including read-in of the redistribution map, has to be considered in this analysis, however, our tests have shown that this does not substantially influence the here presented model run time evaluation.

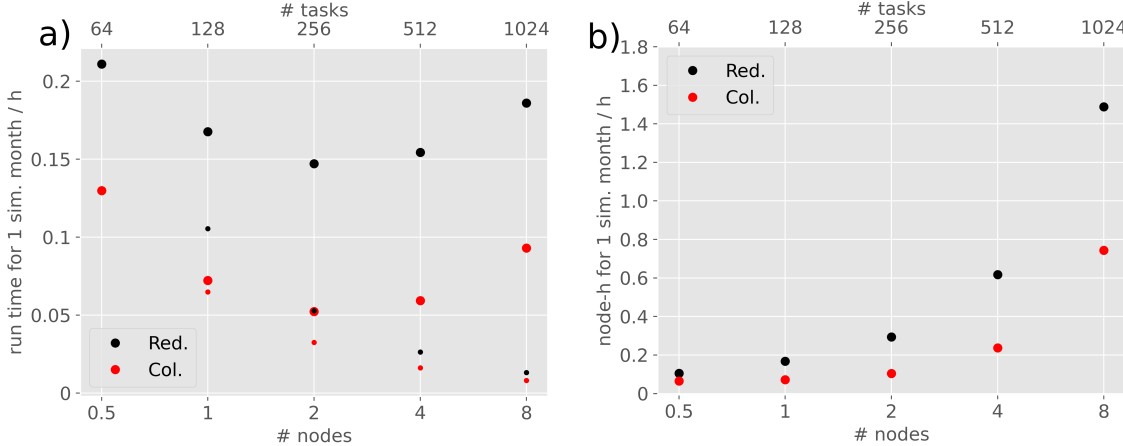

**Figure 7.** a) Run time and b) required node-hours for one simulation month in 5 simulations with (Red. for redistribution) and 5 simulations without the GW redistribution (Col. for columnar). The simulations were performed using 0.5, 2, 4 and 8 compute nodes, respectively. The small dots in panel a) denote the theoretical run time for optimal scaling based on the 64-task simulation.

The run time analysis (Fig. 7a) shows that in both cases, i.e. with and without the GW redistribution, run time decreases with increasing number of used compute tasks until usage of 2 nodes. Between using 2 and 4 nodes, the run time starts increasing
again. This is a computer-specific behaviour and depends on the relation between bandwidth and tasks. In the spectral GCM EMAC, it is mainly the grid point space to spectral space transposition that requires communication between all tasks at each time step. The absolute run time difference between the two simulation types remains similar with increased number of utilised compute tasks. However, in the case with redistribution, run time deviates from optimal scaling more strongly with increasing use of tasks. This is mainly due to the additional all-to-all communication during use of the gl2dc function that was described
in Sect. 3.1. Using only 64 tasks, run time with GW redistribution increases by a factor of around 1.7 (67%) and using 256 tasks (2 nodes) by a factor of 2.8 (184%). As general scaling performance of EMAC diminishes when using even more nodes, the relation improves again in that direction. When we had performed these tests on the HLRE-3 (mistral), which was discarded in year 2022 and featured 36 tasks per node, scaling of standard EMAC did not diminish even when using 16 nodes (576 tasks). On mistral, the run time overhead factor with OGW redistribution was 1.2 (17%) with 1 node and 2.1 (111%) with 16 nodes.
The required node hours that are needed for one simulation month with and without GW redistribution (Fig. 7b) generally shows an increase with increasing number of tasks. As this measure generally represents the run time scaled with the number of nodes, the relations between the two simulations are the same as above. Additionally, excessive memory load on individual compute nodes might become an issue due to the all-to-all communication. We have not faced any problems with regard to this



on the mistral and levante supercomputers at the DKRZ, but especially usage of finer resolution could lead to complications in
that regard depending on the supercomputer architecture.

If we consider that activating the interactive chemistry mechanism MECCA in EMAC (Sander et al., 2019) including numerous tracers, or using the model in a setup with coupled deep ocean can increase the required node hours by more than an order of magnitude, the overhead of the GW redistribution can be regarded as moderate. Hence, for current EMAC this can be a feasible solution for improving the GW representation, however, for highly parallel model designs a different solution must
be sought. Overall, we conclude that chemistry-climate simulations including the GW redistribution with the state-of-the-art EMAC model are affordable and shall be aimed at conducting for a better representation of OGWs (which will be shown in the remainder of this paper). Multi-decadal chemistry-climate simulations with fine enough resolution to disable physical parameterisations such as GWs or convection (which is <1 km horizontal resolution, see Polichtchouk et al., 2022; Kruse et al., 2022) are unlikely in the foreseeable future and other approaches, such as integrated ray-tracers in CCMs are yet to be fully
developed and strongly enhance computing time too. Hence, the GW redistribution can be advantageous for ample time into the future, in which reliable climate projections are key for policy making. However, careful handling of the computing conditions in relation to the redistribution is needed.

## 3.4 GW flux and drag in EMAC test simulations

For a first evaluation of the GW redistribution in EMAC, we conducted test simulations for July 2006 in the T42L90MA
resolution. This is the same period as investigated in the companion paper validating the MWM (Rhode et al., 2023). For this, the greenhouse gases, sea surface temperatures as well as the sea ice concentrations are prescribed. The simulations are purely dynamical, i.e. no chemistry is activated. Fig. 8 shows monthly mean GWMF and zonal GW drag of these test simulations with the standard columnar OGW scheme and with GW redistribution. Here, we chose the redistribution map that was generated particularly for July of 2006 (see Sect. 2.3). We focus mainly on the region around Patagonia and the Antarctic Peninsula, since
OGWs are known to be influenced particularly strongly by lateral propagation there (see e.g. Rapp et al., 2021) and the impact of lateral propagation on dynamics is considered to be large in this region (see e.g. McLandress et al., 2012). For completion, a global depiction of GWMF at the level of GW redistribution is provided in Fig. S4.

The comparison of the OGWMF maps in Fig. 8a and b show that through GW redistribution, GWMF are generally more spread-out. More precisely, there is less GWMF over the continents and over orography, while now there is GWMF over the
oceans and other regions where no or only little orography is located. This agrees with findings in Fig. 2b and the expectations of the implementation and was one of the aims of the parameterisation refinement. GWMF over the ocean is mostly about one order of magnitude smaller than GWMF over land. Most of the GWMF over the ocean can be seen downwind from the Andes and the Antarctic Peninsula, which is consistent with findings from satellite observations (e.g. Ern et al., 2018; Hindley et al., 2020) and high-resolution modelling (e.g. Strube et al., 2021; Polichtchouk et al., 2022). Moreover, some meridional
displacement takes place, which can clearly be seen by comparing panels c and d of Fig. 8. At model level 65, the chosen level of GW redistribution, some of the GWMF is being displaced meridionally and this reduces the GW gap at $60°$S. Figs. 8e and f show that the columnar sum of GW drag displays similar spatial patterns as the GWMF at level 65. This could be expected,





as GW drag is closely related to the location and deposition of GWMF via breaking, which mostly takes place in the middle atmosphere. The comparison of Figs. 8g and h shows that the $60°$S GW gap is partly closed in zonal mean GW drag when GW
redistribution is activated. Further, these figures also display larger zonal mean GW drag in total for the redistributed case, in particular in the upper model layers. This is somewhat surprising and cannot be explained by details of the implementation, as the total GWMF is conserved in our approach. A possible reason are better vertical propagation conditions for the redistributed GWs, but see the following Sect. 3.5 where we discuss this point in more detail.

These results prove the technical applicability of our development and its potential to improve the horizontal distribution
of OGWs in EMAC. However, regarding the redistribution maps from Sect. 2.3, we expected stronger lateral GW displacement through the GW redistribution in zonal as well as in meridional direction. Many of these far away propagating GWs in GROGRAT, however, seem to stem from regions at the continental edges, where no orography is present in the standard EMAC SSO, translating into a lack of GW sourcing in these grid boxes. Therefore, we next evaluate how the updated SSO from Sect. 3.2 influences the results by repeating for Fig. 9 the analysis from Fig. 8 but using the updated SSO.

Comparison of the panels in the left columns of Fig. 8 and 9, respectively, shows that the new SSO generally leads to enhanced launching of GWMF and subsequently stronger GW drag. In particular the continental edges and, in addition, some of the small islands outside the displayed region have larger or rather any GWMF and drag with the new SSO. As expected, this also leads to more far-away transport of the GWMF (Fig. 9b). Considerable patches of GWMF and GW drag can now be seen further away from orography, downwind as far as at $20°$W. Moreover, meridional displacement especially into the latitudes
of the Drake Passage is now represented distinctly (see Figs. 9b and f), such that the $60°$S gap now is completely closed (see Figs. 9d and h). Again, the GW redistribution leads to more total zonal mean GW drag, which will be discussed in the next section (compare Fig. 9g with h). Changes in GW drag directly impact winds and temperatures and indirectly planetary wave propagation (see e.g. Cohen et al., 2014; Eichinger et al., 2020; Šácha et al., 2021), thereby altering stratospheric dynamics. Even differences in GW asymmetry with consistent zonal mean forcing can cause changes in dynamics (see Šácha et al., 2016;
Samtleben et al., 2020; Kuchar et al., 2020). In the next section, we provide a brief analysis of the GW redistribution impact on Antarctic polar vortex dynamics.



**Figure 8.** Comparison of orographic GWMF and GW drag between the standard columnar approach (left) and using the GW redistribution (right) with the standard SSO: a-d) GWMF at model level 65 (level of redistribution) (a,b) and as zonal mean (c,d). e-h) GW drag vertically summed-up (e,f) and as zonal mean (g,h). Note that panels c, d, g and h are depicted in hybrid sigma-pressure model levels to clearly visualise the redistribution level. For reference, the second vertical axis denotes approximated pressure levels and additionally the distribution of hybrid sigma-pressure model levels of the EMAC L90MA setup with pressure altitude is shown in the Supplement.



**Figure 9.** As Fig. 8 but with the updated SSO.





### 3.5 Impacts on stratospheric polar vortex dynamics

To obtain a general picture of the impact of the OGW redistribution on stratospheric Antarctic polar vortex dynamics, we performed 4 time slice simulations over 25 years with sea surface temperatures and sea ice concentration boundary conditions from year 2006 and prescribed GHGs. The 4 simulations are designed in the same manner as the test simulations in Sect. 3.4, with and without GW redistribution and using the new and the old SSO, respectively. The first 5 years of these simulations are considered as spin-up and hence not analysed. The resolution again is T42L90MA and the output time step is 6 h. In contrast to the test simulations above, the redistribution functions used here are based on annual means for year 2006 instead of the July 2006 maps (see Sect. 2.4), but still temporally constant.

To assess these simulations, we first analyse the Southern Hemispheric 20 year climatological zonal mean GWMF at 10 hPa during JJA. Fig. 10 shows OGWMF and total GWMF (OGW+non-orographic (N)GW) for the 4 simulations.

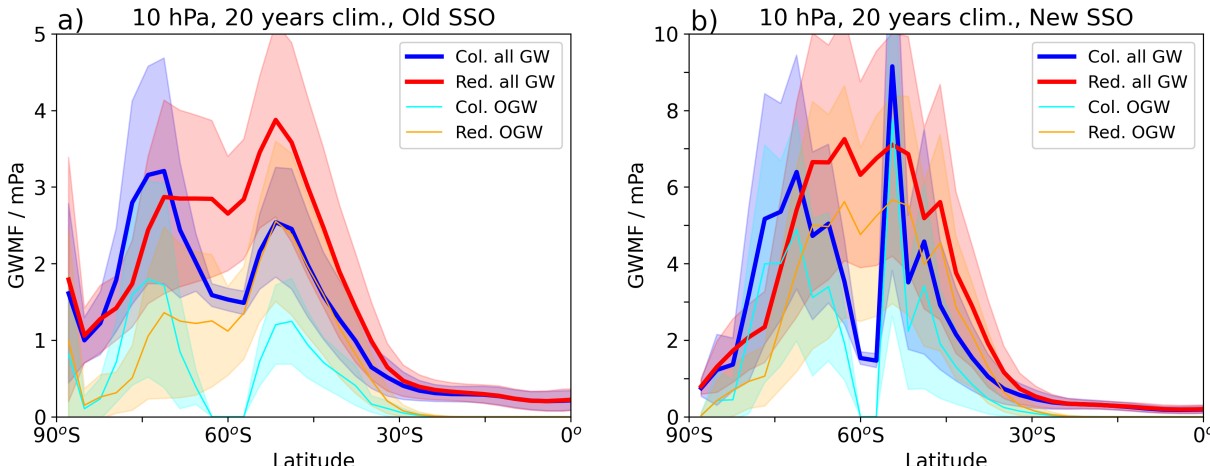

**Figure 10.** Climatological 20 years JJA zonal mean total GWMF and OGWMF at 10 hPa for the simulations with a) the old SSO and b) the new SSO. Col. stands for columnar and Red. for redistribution activated. The shadings denote the variability by means of $1\sigma$ of the GWMF.

The most obvious feature in Fig. 10 is that the GW gap at $60°$S can be closed through OGW redistribution. This could be seen in the previous section already, but now we show that this is the case also for the climatological mean over austral winter, and as conducted using an annual mean redistribution map rather than the specific July map. Using the old SSO (Fig. 10a), the redistribution enhances the total GWMF equatorward of around $70°$S and reduces it poleward of this latitude. In the new SSO case (Fig. 10b), the general picture is similar, but due to more OGW excitation, GWMF is generally larger by about a factor of 2 (note the different scales at the two panels). Moreover, there is a spike with large OGWMF at $\sim50°$S in the columnar-only simulation that exceeds the redistributed case, likely due to GW launching from islands or continental edges. In the redistributed case, enhanced OGWMF can be seen here too, but the spike is smoothed-out through GW redistribution. In general, the accumulated GWMF in the redistributed simulations exceeds the one from the columnar simulations at the





10 hPa level. Consequently, GW drag above 10 hPa is enhanced in the case with GW redistribution (see Fig. S9), consistent with results shown in Sect. 3.4. This behaviour could also be found in a ray-tracing experiment, where the drag increases at altitudes above 40 km and decreases between 25 and 30 km, if horizontal propagation of GWs is permitted (not shown) and

it was also found by Xu et al. (2017). As there is no feedback on the mean flow in the ray-tracing experiments, the changed propagation behaviour cannot be a result of background wind condition modifications through the GW changes. Rather, it must result from different propagation and/or dissipation conditions. One possible explanation could be a systematic GW redistribution to regions with more favourable propagation conditions for the GWs, but a physical reason for such a systematic change cannot directly be established here. What is systematically changing through GW redistribution, however, are the absolute

GWMF values per grid cell at the redistribution level for most regions (see Fig. 9 a,b and Fig. 8 a,b). As our GW redistribution leads to a more spread-out distribution of GWMF, while conserving total GWMF at the redistribution level, it results in lower values per grid cell. Consequently, dissipation from saturation will occur only at higher altitudes (see Equ. 4), explaining why at 10 hPa GWMF is generally larger as a result of GW redistribution. Therefore, GWs generally propagate to higher altitudes through redistribution, and there they exert a stronger drag on the mean flow due to the scaling with inverse density.


The differences in GW drag that the altered GWMF translate to impact the stratospheric dynamics via dissipation. To assess the impact on wind and temperatures, we show in Fig. 11 the climatological JJA zonal mean zonal wind and temperature differences between the two simulations with and the two without redistributed OGWs for old and new SSO.

In the case with the old SSO (Fig. 11a,c), the zonal winds show clear signs of a stronger polar vortex poleward of $60°$S

and a weakened polar vortex equatorward of this latitude, indicating a poleward shift of the polar vortex. These signals might not be statistically significant here, but the monthly mean differences (see Supplement) are partly significant and the signal is consistent from May through September. This indicates that high latitude stratospheric dynamics are clearly influenced by OGW redistribution. These wind changes are generally consistent with a reduction of GWMF at high latitudes and enhanced GWMF north of about $70°$S, as shown in Fig. 10. However, we expect that additional interaction between GWs, planetary

waves and the mean flow (as e.g. analysed by Cohen et al., 2014; Eichinger et al., 2019; Šácha et al., 2021) additionally play a role for the response of the zonal wind to GW redistribution, constituting the well-known non-linearity of stratospheric dynamics. The polar temperature differences in this case, however, are small and barely significant.

In the case with the new SSO (Fig. 11b,d), stronger winds can be seen throughout the polar vortex region when OGWMF is redistributed. This zonal wind signal is partly statistically significant in the individual boreal winter months (see Supplement),

but the temperature signal in the high latitudes is even robust in the seasonal mean differences. The southern high latitude lower stratosphere shows colder air, the higher stratosphere warmer air, and this is the case from April through September. This shows that in the case with the new SSO, the dynamical response to GW redistribution is somewhat different than in the case with the old SSO, and it cannot be explained by altered OGW forcing in the same manner. In principle, the abilities of GWs to propagate to higher levels as discussed above, resulting in reduced drag below around 10 hPa and enhanced drag above is

consistent with the temperature differences, and that again would be consistent with enhanced subsidence at lower altitudes and reduced subsidence above. However, the polar vortex strength increases also at levels above 10 hPa, where the total drag





by OGWs is strongly enhanced (see Fig. S9). This again emphasises the non-linear nature of the response of high-latitude stratospheric dynamics to forcings, and an in-depth analysis of feedbacks between GWs, planetary waves and the mean flow would be necessary to gain conclusive understanding of the modelled response to the GW redistribution. This includes its

sensitivity on the background state (which Sigmond and Scinocca, 2010, showed to control wave propagation conditions and its sensitivity to changes) as indicated by the differences between the response in the new and old SSO case.

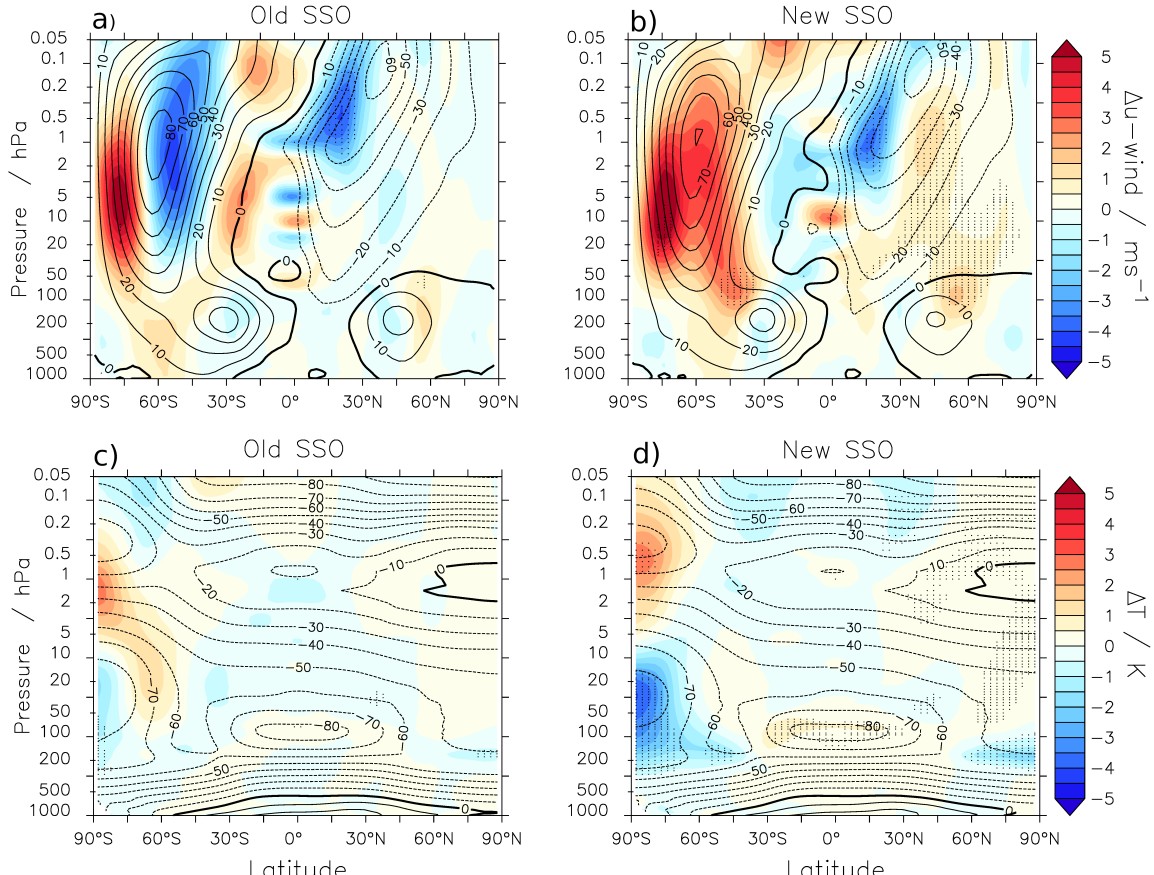

**Figure 11.** Zonal mean zonal wind (a and b) and temperature (c and d) differences in JJA between the simulations with and without OGW redistribution using the old (a and c) and the new (b and d) SSO. The contour lines show the climatologies without OGW redistribution in $ms^{-1}$ and $^oC$, respectively. Dotted regions depict where the differences are statistically significant on the 95% level.

Another property of the Antarctic polar vortex that is of high interest is the day of year of its breakdown (final warming). The persistence of the polar vortex through spring is relevant because of both ozone chemistry and its impact on tropospheric dynamics. In Fig. 12 we show the distribution of the stratospheric final warming day of year for the 4 simulations. Following

for example Black and McDaniel (2007) and de la Cámara et al. (2016), the final warming day of year is diagnosed by using 5-day running means of daily data to calculate the final warming date as the last time that the zonal-mean zonal westerly wind at $60°S$ drops below $10\,ms^{-1}$ until the subsequent autumn. Fig. 12 shows that GW redistribution slightly shifts the mean of





the final warming date to later times in the year, though barely statistically significant. With the old SSO, the median is shifted from day 298 to day 301 and with the new SSO from day 298 to day 303. In contrast to many other models, in EMAC the

Antarctic polar vortex tends to break down too early (Jöckel et al., 2016) compared to reanalysis data (e.g. de la Cámara et al., 2016, found day ∼315 to be the median in ERA-Interim), thus the redistribution acts towards improving the model in this regard. Note that also the structures of the final warming day distributions change, in general towards a larger spread of the final warming date. These results are in line with the study by Gupta et al. (2021) who state that parameterised GW drag at 60°S generally provides more than half of the wind deceleration for the Antarctic polar vortex breakdown, occasionally the

amount can even be as high as the total necessary deceleration. Note further, that the generally stronger OGWMF with the new SSO compared to the old one alters the final warming day distribution, but not its median value. As before, it will be of interest to study the interaction between GWs, planetary waves and the mean flow to more thoroughly understand this behaviour.

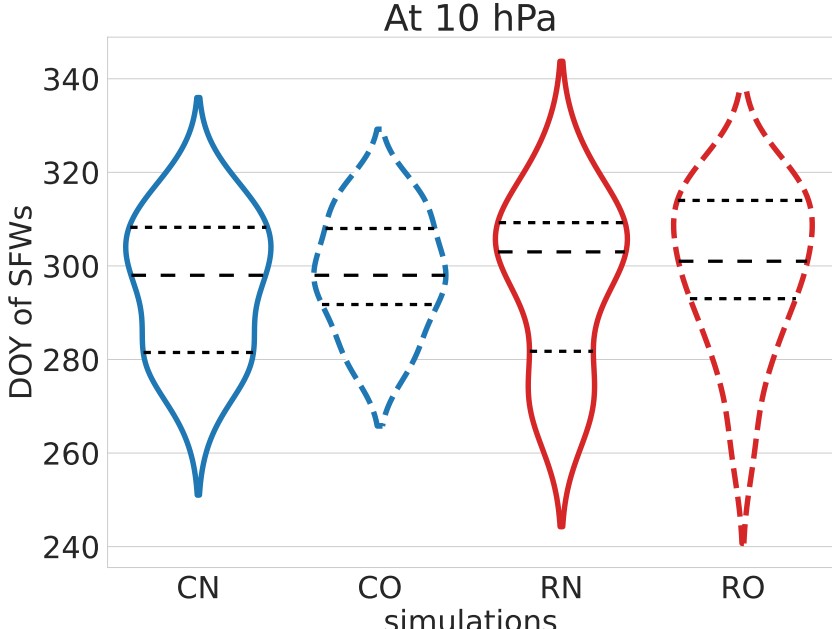

**Figure 12.** Distribution of day of year (DOY) of the final stratospheric warming for JJA in the 20 year time slice simulations. The different simulations are denoted by C for columnar, R for redistribution, N for new SSO and O for old SSO. The horizontal dashed lines mark the quartiles.

For additional investigation of the Antarctic polar vortices in the 4 simulations, we use geometric-based methods by taking 2-dimensional moments of dynamical fields. These diagnostics were for example put forward by Mitchell et al. (2011) and Se-

viour et al. (2013) and include the vortex excess kurtosis, aspect ratio and centroid latitude. The excess kurtosis is a non-linear and state-dependent diagnostic (see Matthewman and Esler, 2011), it is a measure of the departure of the vortex shape from





an ellipse (Scott, 2016). Positive kurtosis values correspond to an eye shape with the vortex narrowing at its extremities and larger positive values could be associated with higher probability of filamentation events of vortex material into the surf zone (Mitchell et al., 2012; Scott, 2016). GW redistribution enhances the kurtosis median at 10 hPa by around 10% in both cases
(see Fig. S10 for distribution). This can mean a significant change in circulation, but in detail analyses need to be conducted for drawing clear conclusions here. The vortex aspect ratio median and distribution at 10 hPa is clearly changed through GW redistribution in the case with the new SSO, but not in the case with the old SSO (Fig. S11). A change in vortex form is in line with Fig. 11b and d. The centroid latitude does not reveal a clear picture as to any systematic changes through GW redistribution in our simulations (Fig. S12).


It has to be noted that some of the differences in the results are stronger between the (columnar approach) simulations with different SSOs than with or without redistribution. As the SSOs strongly alter GW launching and therefore GW drag, this is not surprising, however, as this measure does not change the spatial OGW distribution, which has been shown to be important for planetary-scale wave fields (Šácha et al., 2016, 2021; Samtleben et al., 2019, 2020), this does not help to
simulate GW drag closer to reality and this way, GWMF cannot be constraint to observations. In general, these results should be interpreted with caution, as the model has not been retuned. The scientific meaning of the results is therefore rather of qualitative nature, as to how much potential for changes in large-scale dynamics the new developments actually have. It remains to be shown if these signals are robust in various simulations with modified parameters and also with various horizontal and vertical resolutions. Different redistribution maps can be applied, the redistribution altitude can be varied, possibly even implemented
dynamically, or the regular tuning parameters of the OGW scheme customised. For this, a concrete idea for tuning these new model developments needs to be applied and we lay out a plan for this in the following section.

## 4    Summary and Discussion

In this paper, we describe a simplified solution of a 3-dimensional orographic gravity wave (OGW) representation in the global chemistry-climate model (CCM) EMAC (Jöckel et al., 2010). The lack of horizontal OGW propagation constitutes a long-
standing problem in atmospheric modelling, potentially with substantial impacts on middle atmospheric dynamics that are crucial for accurate climate projections (McLandress et al., 2012; Geller et al., 2013).

As horizontal communication in general circulation models (GCMs) is computationally expensive, the idea is to implement a cost-efficient emulation of lateral OGW propagation. This is realised by redistributing orographic GWMF at one single altitude by use of tailor-made redistribution maps. The ray-tracing model GROGRAT coupled to a mountain ridge parameterisation
(see Rhode et al., 2023) is applied to generate these maps. The latter are 4-dimensional probability functions describing GW redistribution via horizontal propagation as a mapping from a source latitude-longitude grid to a target latitude-longitude grid (therefore the four dimensions are source-latitude, -longitude and target-latitude, -longitude). The individual map value depends on the relative fraction of the total GWMF in the given source cell that has been transported to the corresponding target grid box. The target altitude is one of the free parameters of the redistribution and was determined here to provide best results for



40 km, which results in an optimal approximation of horizontal GW drag redistribution. The ideal altitude for GW redistribtion
in the CCM was determined to be at 15 km altitude, corresponding to around 120 hPa. For this, the GW redistribution was
inferred from the ray-tracing model data and the deviation from the setup with full 4-dimensional (3 spatial dimensions and
time) propagation in the monthly mean was analysed. Both redistribution and target altitude can be subject of further studies
and in practice might be adjusted depending on the particular question of the given study, but here, we aim at seeking the most

general solution. Additionally, the redistribution maps can be generated using annual means or monthly means of particular
years, or of averages over several years and the meteorological boundary conditions can be taken from reanalyses or from
model simulations. All these different options should, for now, be chosen depending on the science question(s) to be answered.
It is worth to mention, however, that a good approximation for individual months can already be gained from an annual mean
redistribution pattern. Therefore, a major part of the horizontal GW propagation can be described by a general pattern. In the

long run, a flow-dependent redistribution function might be sought, in order to provide full flexibility of the model and higher
accuracy in the propagation approximation. This could be realised by analysing the dominant modes of the time-varying
redistribution function, and their relation to variability in the large-scale flow. Possibly, the redistribution in a certain region, or
even in a hemisphere, can be described by few (up to 3) leading modes. Online calculation in the GCM of the weights for the
superposition of the dominant redistribution mode could then be the basis for the flow-dependent GW redistribution.

The implementation in the CCM EMAC works such that GW redistribution is applied at a specific model level, which can
freely be chosen. To be closest to 15 km altitude with this, we here chose model level 65 for the used L90MA setup. Below
the GW redistribution level, EMAC still comprises the ordinary columnar OGW scheme. Above this level, the redistributed
GWMF again is treated as in the columnar approach, but now partly in other columns. The basis for this is the standard OGW
parameterisation in EMAC, i.e. the scheme that was initially implemented by Lott and Miller (1997) and Lott (1999). As in

this scheme, several incident quantities and parameters from the subgrid-scale orography (SSO) are required for calculating
the saturated GWMF after wave breaking, an additional variable ($Z_{oro}$) comprising these quantities has to be redistributed.
After performing the horizontal redistribution of the GWMF on global fields by looping over the blocks of the decomposition,
transposition of these fields to the decomposition is performed. This allows usage of the redistributed fields in output and in the
next time step and makes the implementation independent of the simulation-specific decomposition. Amemiya and Sato (2016)

implemented 3-dimensional propagation into the GCM MIROC (Model for Interdisciplinary Research on Climate; Watanabe
et al., 2008) by means of including discretised ray-tracing equations. To our knowledge, however, this implementation only
works in a specific decomposition and may hence not be suitable for operational use in CCM simulations.

Our analyses show that the computational overhead of the GW redistribution in EMAC is moderate. The main factor increas-
ing communication time and consumption of node hours is the additional transposition of the global fields to the decomposition,

where the entire information of the redistribution has to be present on each processing element. This enhances the overhead
with increasing usage of computing tasks, such that a factor of 2-3 in node hours requirement is reached when the model is
run on 200 to 500 compute tasks. This means that for highly parallel model designs, this approach is not feasible. For that, a
model integrated ray-tracer with a communication approach between the neighbouring grid boxes or simply running the model
at GW allowing resolution might be more practicable strategies. At present, however, such approaches are far too costly for





multiple decadal or centennial simulations (in particular when including comprehensive chemistry) and computing resources (computing and data storage facilities) are not growing quickly enough to provide such possibilities in the near future. This is particularly true, as for climate projections it is not one 'true' simulation that is needed, but instead an ensemble of simulations with different settings, models and scenarios. For current EMAC chemistry-climate simulations, which are usually run on around 400-500 tasks on state-of-the-art high performance computing systems, the GW redistribution as presented here is

absolutely viable. In particular, in comprehensive CCM simulations (including the chemical mechanism, numerous tracers and possibly a coupled ocean model) the GW redistribution overhead will be a minor part of the computing time consumption.

  First EMAC test simulations with this new implementation show that the GWMF is generally more spread-out and also extends over the ocean when the GW redistribution is activated. In particular, some GWMF is located downwind from the Andes and the Antarctic Peninsula, as expected from high-resolution modelling (Strube et al., 2021; Polichtchouk et al., 2022)

and satellite observations (Ern et al., 2018; Hindley et al., 2020). Moreover, meridional displacement of the GWMF at the level of redistribution to some degree closes the well-known GW gap at $60°$S in the CCM. Our simulations show a larger total zonal mean GW drag above $\sim 10\,\mathrm{hPa}$ when the GW redistribution is active and less GW drag below. By implementation design, the total GWMF is globally conserved when applying the GW redistribution, this is hence an indirect effect, which is also reproduced in ray-tracing experiments. Most likely, the propagation to higher altitudes results from lower values of GWMF per

grid cell, so that saturation is reached only at higher levels.

  We also show that the GWMF and drag results improved considerably by using an updated SSO, especially because GWMF that is zonally transported far-away from its source was underrepresented with the standard SSO. The reason is that in GROGRAT, particularly GWs that are excited at the continental edges experience strong lateral propagation and there the standard EMAC SSO does not feature any mountains. With the new SSO, also meridional displacement increases, leading to a complete

closure of the GW gap at $60°$S.

  The OGW redistribution bears impacts on the representation of stratospheric dynamics, in particular on the Antarctic polar vortex with implications for Antarctic stratospheric ozone and its trends. The results of Amemiya and Sato (2016) also show an alleviation of the $60°$S gap. However, they could not find considerable changes in the polar night jet, possibly due to compensation mechanisms with planetary waves (see e.g. Cohen et al., 2014; Sigmond and Shepherd, 2014; Eichinger et al., 2020). Our

simulations in contrast, show that significant differences in structural zonal mean zonal wind patterns can be generated through the OGW redistribution and this can translate into robust high latitude temperature changes. In contrast to most CCMs, EMAC comprises a warm pole bias in the Southern Hemisphere (Jöckel et al., 2016), which means that the here simulated cooling in the lower stratosphere through OGW redistribution signifies an improvement of this bias, with the potential to advance the representation of polar ozone. Also a shift of the final Antarctic polar vortex breakdown to later days in the year could be detected

in the simulations, which corresponds to results by Gupta et al. (2021) who found a large contribution of parameterised GW drag at $60°$S for the wind deceleration at the end of austral winter in ERA5 data. Moreover, specific diagnostics of the vortex geometry, i.e. excess kurtosis, aspect ratio and centroid latitude, reveals that clear changes of the polar vortex geometry occur, which has implications for its stability and thereby stratospheric warming frequency. In further consequence, this can have large impacts on the simulation of stratospheric polar ozone in the model, i.e. the representation of the ozone hole, and the sim-



ulated ozone recovery across the 21st century. Therefore, it should be of high interest to consolidate these new developments for further use in simulations with enabled interactive chemistry to perform and analyse climate projection simulations with redistributed GWMF. This could include analysis of the GW redistribution impacts in a changing climate concerning investigations of GW hotspots, stratospheric warmings, polar vortex strength and variability, ozone chemistry, downward coupling and interactions between GWs and planetary waves on various temporal and spatial scales.

Refinements of the GW parameterisations in GCMs are increasingly important for climate modelling. Partly, because more and more knowledge is still being gathered about GWs that needs to find its way into the schemes (see Plougonven et al., 2020). But also, as the resolution of model simulations tends to be increasing, more (but not all) of the GW spectrum becomes resolved and the parameterisations have to be revised. On that account, it will become even more important to be aware of the possibilities lingering in the depth of the schemes and to have the parameterisations as close to the physics as possible. For

this, testing the GW redistribution in other horizontal and vertical resolutions in order to assess their resolution-dependency and whether the modifications have to be designed in scale-aware manners will be important. Moreover, revisiting the above described free parameters of the redistribution (and analysing how the impact of redistribution varies with climate change) and of the GW scheme itself will be an interesting task. On that note, also the importance of a second, or a dynamical, altitude of redistribution should be assessed. However, the next step for consolidation of this development, will have to be elaborate model

tuning. This could now be conducted in a novel manner, i.e. not by using the GW scheme to tune the model to a certain climatic state, but by directly constraining the GWMF to those of high-resolution simulations such as conducted by Polichtchouk et al. (2022), to satellite observations such as shown in Ern et al. (2018) and Hindley et al. (2020), as well as to high resolution point observations as for example by Kaifler and Kaifler (2021) and Reichert et al. (2021). This would imply a new relevance for GW schemes and allow more meaningful comparisons with observations. One important question of this will be if horizontal

GW propagation is the reason for model deficiencies in representing the Antarctic polar vortex and associated ozone chemistry, which can then lead to more confidence in simulating climate change across the upcoming decades.



*Author contributions.* RE, SR, HG and PPr designed the study. RE and SR wrote the paper. SR generated the redistribution maps, the GROGRAT analyses and the new SSO with support by PPr, RE and HG. RE implemented the redistribution in EMAC with help by PJ, AKe, BK and HG, conducted the GCM simulations and analysed the results with support by AKu, HG and PPi. All authors helped with discussions
and with revising the manuscript.

*Data availability.* The EMAC simulation results are archived at DKRZ, these output data as well as those from the Mountain Wave Model and GROGRAT simulations are available upon request to the authors.

*Code availability.* The Modular Earth Submodel System (MESSy) is continuously further developed and applied by a consortium of institutions. The usage of MESSy and access to the source code is licenced to all affiliates of institutions which are
members of the MESSy Consortium. Institutions can become a member of the MESSy Consortium by signing the MESSy Memorandum of Understanding. More information can be found on the MESSy Consortium Website (http://www.messy-interface.org). The code presented here has been based on MESSy version 2.55.2 and will be available in the next official release (version 2.56). Access to the code used in the Mountain Wave Model and GROGRAT is possible upon request to the authors.

*Competing interests.* The authors declare that they have no conflict of interest.

*Acknowledgements.* The authors want to thank Inna Polichtchouk, Petr Sácha and Chris Kruse for discussions and Robert Reichert for valuable comments on the manuscript. Moreover, we acknowledge usage of resources from the Deutsches Klimarechenzentrum (DKRZ, German Climate Computing Centre) granted by its Scientific Steering Committee (WLA) under project ID bd1022.

*Financial support.* This study was funded by the German Ministry for Education and Research (grant no. 01 LG 1907; project WASCLIM)
as part of the Role of the Middle Atmosphere in Climate (ROMIC) programme and by the Grantová Agentura Ceské Republiky under Grant Nos. 21-20293J. AKu acknowledges support from Deutsche Forschungsgemeinschaft under grant JA836/43-1.



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
