# Peer review of "Emulating lateral gravity wave propagation in a global chemistry-climate model (EMAC v2.55.2) through horizontal flux redistribution"

_EGUsphere, 2023_

## Author Comment (AC1)

On:

**"Emulating lateral gravity wave propagation in a global chemistry-climate model (EMAC v2.55.2) through horizontal flux redistribution' GMD, 2023, by Eichinger et al..**

August 10, 2023
* * *
**1 Reply to Review #1**

Dear Anonymous Referee #1,

thank you for your time and effort to review our paper. Please find our answers (in italic) to the points of your revision (in bold) below.

Best wishes
Roland Eichinger and Sebastian Rhode on behalf of all authors

1. **Why are MWs assumed to be perpendicular to the source ridge? In reality, the winds can be oblique to the ridge.**

   *Within the Mountain Wave Model (MWM), we consider mountain ridges to be more-or-less two-dimensional. Studies of the GWs excited by such mountain ranges have shown that the phase fronts align with the mountain ridge even for barely anisotropic ridges (i.e., 3:1 length vs. width) and, therefore, the GWs are launched with the wave vector perpendicular to the ridge and not necessarily opposite to the wind. The wind direction, however, impacts the amplitude of the explicit mountain wave (MW) since only the wind perpendicular to the mountain ridge excites GWs. One of the first studies of this effect has*

been done by Hines [1988]. Subsequently, **perpendicular** wave vectors have been used in (almost) all studies on the topic of MWs [e.g. Bacmeister, 1993] and can be seen in observations and model data as well [e.g. Kruse et al., 2022].

The text has been altered as follows to be more clear: "For initialisation, MWs are assumed to always launch perpendicular to the source ridge**, which is in line with previous studies on MWs [e.g. Hines, 1988, Bacmeister, 1993] and can be seen in observations and model data as well [e.g. Kruse et al., 2022].**"

2. **L136: The resolution of ERA5 is 0.25 degree, right?**

*Yes, right, we thank the reviewer for spotting this mistake, we corrected it.*

3. **Do you mean there are many ridges within a grid cell? I'm confused about this because, in OGWD parameterization, only a dominant ridge is considered within one grid cell.**

*We assume this comment refers to lines 170-171. In that regard, the MWM works differently than the OGW parameterisation, where only one individual ridge is assumed. In the MWM, there are many ridges with various orientations within a single model grid cell, and for all of these, different propagation patterns are calculated according to their wave properties. This leads to a wide spread of resulting GWMF. To generate the redistribution maps, all ridges from the ridge parameterisation that launch GWs within a grid cell are averaged over, and all GWMF in a target grid cell is averaged over. We made this more clear by the following modification of the text:*
*"But also the relatively large grid size used here ($\sim$2.8°$\times$2.8°) leads to the inclusion of many differently oriented ridges in one cell **within the MWM consideration**."*
*and by adding the following sentence to line 144:*
*"**The grid cell size can be chosen on individual needs and determines how many individual MWs, and therefore GWMF, are averaged over in the source as well as in the target grid cells.**"*

4. **Remove "it was used" which appears to be redundant.**

*The 'it was used' is not redundant, it is needed to specify the configuration we had used here. But we understand that you do not like the sentence as it is and therefore rephrased it to read:*

***"Thirdly, we use the MWM allowing full lateral propagation of GWs, i.e., in the same configuration that was also used to generate the redistribution maps."***

5. **About Figure 3. Firstly, why the GWMF of NO_HOR is less than those redistributed ones in the lower altitudes (especially below 10 km)? Wouldn't it be greater in the absence of lateral propagation of OGWs? Secondly, why are the GWMFs different for different $H_{tar}$? Taking $H_{tar} = 40$ km and 45 km for example, they should be the same below 40 km since no lateral propagation below this level for both cases, right?**

*Figure 3 shows the average **deviation** of the corresponding sensitivity simulation to the (reference) simulation with full propagation at each altitude level. In the sensitivity simulations with GW redistribution, the redistribution has been performed at the lowermost level, i.e. at the surface. Therefore, at lower altitudes, the emulated propagation through one-time GW redistribution is overestimating lateral propagation, and this leads to a stronger deviation to the reference simulation (where the GWs have not yet propagated as far) than the non-propagating data, NO_HOR. In that sense, the NO_HOR run performs better at describing the propagation at the lowest altitudes. At higher altitudes, however, the purely columnar GW treatment leads to a far greater deviation from the reference simulation than with redistribution.*

*The deviation for different target heights, $H_{\mathrm{tar}}$, differs (also below $H_{\mathrm{tar}}$) because the corresponding redistribution maps are different and redistribution takes place at the lowest level. We want to emphasize again (as in Sect. 1, Sect. 2.3, Fig. 1 and at the beginning of Sect. 2.4) that the redistribution depends on two different parameters: the target height, $H_{\mathrm{tar}}$, and the redistribution height, $H_{\mathrm{rd}}$. $H_{\mathrm{tar}}$ defines the altitude at which the GW location is taken for generating the redistribution map from the MWM simulation with full propagation configuration. $H_{\mathrm{tar}}$, therefore, has a direct impact on the redistribution map. On the other hand, $H_{\mathrm{rd}}$ is the altitude at which the one-time redistribution is performed and has no impact on the redistribution map itself. Therefore, the GWMF after redistribution with different maps*

*(generated with different $H_\mathrm{tar}$) should be different at all altitudes above the redistribution height $H_\mathrm{rd}$. In panels a, b and d, e of Fig. 3, we only consider the effect of $H_\mathrm{tar}$ and fixed $H_\mathrm{rd}$ to the lowermost altitude level. The profiles are thus different at all shown levels.*

*To make these points more clear, we added the following sentences to the manuscript:*

*– "**Note that for now, we are only considering a variable target height, $H_\mathrm{tar}$, to assess the ideal $H_\mathrm{tar}$ for generating the redistribution maps. An optimised value for the redistribution height, $H_\mathrm{rd}$, will be estimated later on in this section.**"*

*– "**Here the redistribution height is fixed to the lowermost level. Therefore, all data sets are different for all levels due to the different redistribution maps for each case.**"*

*Moreover, the sentence*
*"For the considerations in Fig. 3, the redistribution height was chosen case specific at the cross-over point, i.e. the height, above which a redistribution yields a net positive improvement"*
*has been changed to*
*"**In principle, such an optimal redistribution height could be found for any specific case and would vary depending on season and target height, see e.g., the differing cross-over points (the height above which a redistribution yields a net positive improvement) for the annual and monthly consideration in Fig. 3.**".*

*and the figure caption has been modified to now read:*
***Comparison of MWM simulations with GW redistribution (with varying $H_\mathrm{tar}$, coloured lines) and vertical-only propagating MWM simulations (NO_HOR, black lines) to a simulation including horizontal propagation (REF) for a) and d) GWMF and b) and e) GW drag. To show the effect of the target height, $H_\mathrm{tar}$, the GW redistribution is performed at the lowermost level in these 4 panels, leading to profile differences at all altitudes. The horizontal axis shows the relative deviation to REF at the corresponding altitude in monthly and global mean. Panels c) and f) show the maximum improvement (reduction of deviation to REF) for different $H_\mathrm{tar}$ with optimal redistribution height (i.e., the redistribution takes place at the cross-over of the black and the respective coloured curve). This is proportional to the area between the black and the coloured curves in panels a and b (only to the left of the black curve). The redistribution maps were generated from simulations of July 2006 in panels a-c***

*and from simulations of the entire year 2006 in panels d-f and applied to simulations of July 2006 in both cases.*

6. **About the implementation in L264-273. When applying the lateral propagation, how far can the parameterized OGWs propagate in the horizontal direction? In other words, is there an upper limit of horizontal distance for the lateral propagation of OGWs? Moreover, can you talk about the potential influence of applying the redistribution only at one single altitude level? Clearly, the OGWs propagate more and more laterally with height. Applying at single level means omitting the lateral propagation below this level and underestimation above this level. Is it possible to apply the lateral propagation at full altitudes? In this case, Zoro is not a dynamic parameter (L309) and you don't have to calculate Eq. (5), right?**

   *By design, there is no limit in how far the GW flux can be redistributed horizontally. To make this more clear, we add the following to line 267:*
   *"... to apply the **global** GW redistribution ... "*
   *Below the level of redistribution, lateral GW propagation is (still) omitted in our approach. Above the level of redistribution, we make a compromise for lower and higher altitudes, which is not only determined by the level of redistribution but also by the target height (see Sect. 2.4). So it is not correct to simply say that we have an underestimation of lateral propagation, the reviewer is referred to Sect. 2.4 where this topic is described and analysed in detail.*
   *We assume that with "full altitudes" the reviewer means "all" altitude levels. Theoretically, this is possible, but then we would need redistribution maps for each altitude and the computing time overhead in the GCM would be tremendous. For such an approach, it would be more sensible to integrate a ray-tracer into a GCM (which we discuss in L572), then you could also pass on wave properties and Zoro is not needed. However, in the present study, we explicitly sought for a simplified and computing-cost-efficient solution.*

7. **Why the map is 4D? Is it time-varying and spatially different?**

   *The redistribution maps are 4-dimensional because they have two latitudes and two longitudes (for source and target). As discussed, temporal variation is a possible option for the future. No line number was added to this comment, but we assume the reviewer means line*

*294, as this is the only place where we mention the redistribution map dimensions without specifying this. To prevent this confusion from happening, we changed the sentence to:*
*"The 4-dimensional redistribution map (2 **latitudes and 2 longitudes** plus a possible time dimension) is ...."*

8. **Ok, the authors stated that the map is temporally constant. How about using a time-varying map, at least monthly variation? (see the notably monthly variation in Fig.4a)**

*Yes, the GW redistribution map we use here is temporally constant, but technically time-varying maps could be used, and including monthly variations of the maps might lead to further improvements. We do discuss this in the discussion section, please see lines 545-554. To make it clearer, we add*
*"Additionally, the redistribution maps can be **used time-varying**, generated **by** using annual means or monthly means of particular years, or ...."*
*here. Our results show that the annual mean GW redistribution map already reproduces the leading wave propagation mode (548-549), so the main model improvement can already be achieved with this simpler solution. However, for further developments and studies, the technical basis for time-varying GW redistribution maps is now there. Please also note that a monthly varying GW redistribution map would for example not capture the changes through polar vortex breakdown and associated refraction index changes within one month. Hence, we also discuss the option of a flow-dependent redistribution function, that might be able to be realised through few leading modes only (L549-553).*

9. **Is it possible that the difference between the total GW drag may be due to the different wind circulations in these two experiments which determine the wave source and breaking?**

*We assume the reviewer refers to lines 445-459 (and 589-590). We have ruled out this possible explanation because we found the behaviour also in the ray-tracer, and there the wind conditions were the same (i.e. prescribed) in both simulations (see lines 450-451). But also the fact that the behaviour is found systematically in both model setups with new and old SSO, as well as the magnitude of the changes that is beyond variability, clearly speak against this hypothesis. Therefore, we are convinced that our proposed explanation (lower values of GWMF per grid cell, so that saturation is reached only at higher levels) is much*

*more reasonable.*, but see also our replies to reviewer#2 who brought up a second possible explanation, which we included in the text now.

**References**

J. T. Bacmeister. Mountain-wave drag in the stratosphere and mesosphere inferred from observed winds and a simple mountain-wave parameterization scheme. *J. Atmos. Sci.*, 50:377–399, 1993.

C. O. Hines. A modeling of atmospheric gravity waves and wave drag generated by isotropic and anisotropic terrain. *Journal of Atmospheric Sciences*, 45(2):309 – 322, 1988. doi: https://doi.org/10.1175/1520-0469(1988)045¡0309:AMOAGW¿2.0.CO;2.

C. G. Kruse, M. J. Alexander, L. Hoffmann, A. van Niekerk, I. Polichtchouk, J. Bacmeister, L. Holt, R. Plougonven, P. Sacha, C. Wright, K. Sato, R. Shibuya, S. Gisinger, M. Ern, C. Meyer, , and O. Stein. Observed and modeled mountain waves from the surface to the mesosphere near the Drake Passage. *J. Atmos. Sci.*, pages 909–932, 2022. doi: 10.1175/JAS-D-21-0252.1.

---

## Author Comment (AC2)

On:

**"Emulating lateral gravity wave propagation in a global chemistry-climate model (EMAC v2.55.2) through horizontal flux redistribution' GMD, 2023, by Eichinger et al..**

August 10, 2023
* * *
**1 Reply to Review #2**

Dear Anonymous Referee #2,

thank you for your time and effort to review our paper. Please find our answers (in italic) to the points of your revision (in bold) below.

Best wishes
Roland Eichinger and Sebastian Rhode, on behalf of all authors

1. **Section 2.1: What is the reason of using idealized, Gaussian-shaped mountain ridges for the MW parameterization?**

   *We use idealised, Gaussian-shaped mountain ridges as these are often-times the object of study in MW generation investigations [e.g. Lott et al., 2020, 2021]. This allows for a straightforward conversion of mountain parameters, i.e., width and height, to the initial GW parameters, i.e., wavelength and amplitude. By using these sets of idealised mountains, the MW sources can be localised in the orography.*
   *For clarification, the following sentence has been added to the manuscript:*
   *"The Gaussian mountain shape has been used in many MW studies and*

*allows straightforward conversion of ridge parameters, i.e., width and height, to the initial GW parameters, i.e., wavelength and amplitudes."*

2. **Eq 3. Just a clarification, the terminology may have confused me: the momentum flux $\tau_{m1}$ that is being redistributed is taken at the model level 65, right below 15 km. And this model level is labeled "src" in eq (3)? So "src" is not the GWMF at source level in the parameterization, but at the chosen level for redistribution?**

   *Yes, right, all this refers to GW flux at the level of redistribution, not at the GW source level. To make this more clear and avoid any confusion, we slightly adapted the explaining sentence there to read:*
   *".... and the subscripts tar and src denote **(horizontal)** target and source **grid cell** of the GWs **at the level of redistribution, respectively**.*

3. **Figs. 8 and 9, Lines 405 and 450: Regarding the increased drag at upper levels with redistributed flux, part of the reason of this behavior might indeed be due to more favorable vertical propagation conditions around the polar night jet. If a fraction of the momentum flux generated at the Andes and the Antarctic Peninsula is redistributed around 60S, where the zonal mean wind maximum is located, the saturated flux given by eq. (4) would be larger, hence allowing the waves to propagate upwards without dissipating. Does this make sense? Plougonven et al. (2017) reported a tendency in observations and high resolution simulations for large momentum fluxes to be located at the jet maximum, which was explained in terms of horizontal propagation.**

   *Thank you for this interesting theory. This makes sense to us and we have now included it in the explanation part in section 3.5. This part now reads:*
   *"... regions with more favourable propagation conditions for the GWs. A physical reason for this could be that around $60°S$, where the zonal mean wind maximum is located, the saturated GWMF given by Eq. 4 is larger, hence allowing GWs to propagate upwards without dissipating. This is supported by Plougonven (2017), who reported large GWMF to be located at the jet maximum in observations and high-resolution simulations. Another systematic change through GW redistribution are the absolute..."*

*Moreover, we added it to the Summary stating:*
*"... and from more favorable vertical propagation conditions around the polar night jet where some GWMF has been redistributed to."*

4. **Fig. 11. I would suggest to add some panels to this figure showing the comparison with ERA5, this would be valuable to assess whether the implemented redistribution works in the right direction, with all the caveats regarding the lack of refined tuning.**

   *Thank you for the suggestion, but we refrain from adding these panels. This is certainly a good idea for a follow-up study, but it opens a can we do not want to open in this one. The caveats you mention are too many to deal with here. Using the two different SSOs already leads to unclear results as to how the vortex is altered through the OGW modifications. This means, that already here, we receive ambiguous results and cannot clearly state how exactly the dynamics will change. As further work on this will require large tuning efforts, an eye will have to be kept on stratospheric dynamics and systematic analyses will allow clear statements. For now, however, we want to leave it at the point where we show that the OGW redistribution has the potential for significant changes of polar vortex dynamics and based on what we know from previous EMAC model evaluations, they point into a good direction, although not as clearly as one would have wished for.*

5. **Although the interaction between the modified GW drag, planetary wave driving and the mean circulation well deserves a separate study, it would be very interesting to briefly analyze changes in planetary wave driving in these 4 EMAC runs. I would suggest to add the corresponding latitudinal distribution of WP flux divergence to the panels in Fig. 10. According to Garcia et al (2017), there is a strong compensation between GW drag and resolved forcing around 60S due to the columnar approach followed by orographic GW parameterizations. Besides, these plot may help explain to a first order the changes in the zonal mean zonal winds and temperatures given in Fig. 11.**

   *To meet your point, we added a figure to the supplement displaying zonal mean differences of OGWD, NGWD and EPfd (see also below). Moreover, we added to the text:*
   *"As noted before, OGW drag strongly increases in the upper strato-*

[Figure]

Figure 1: Zonal mean difference of (a, d) orographic gravity wave drag, (b, e) non-orographic gravity wave drag and (c, f) Eliassen-Palm flux divergence between simulations with OGW redistribution and with columnar OGW approach for (a-c) new SSO and (d-f) old SSO. The contours show the respective climatology of the columnar approach simulation and stippled regions show where the differences are signficant on the 95% level.

> sphere. This increase is partly compensated by a decrease in non-orographic GW drag, and partly by planetary wave drag. However, we do not find a systematic compensation of the (missing) drag at $60°S$ as reported by Garcia et al. (2017). As shown by Eichinger et al. (2020), the occurrence of compensation and thereby also the impact on zonal winds seems to strongly depend on the basic state, and in cases also amplifying effects are found. An in-depth investigation of the wave-wave and wave-mean flow interactions will be needed to determine what exactly are the crucial mechanisms here."

**References**

F. Lott, B. Deremble, and C. Soufflet. Mountain waves produced by a stratified boundary layer flow. part i: Hydrostatic case. *Journal of the Atmospheric Sciences*, 77(5):1683 – 1697, 2020. doi: https://doi.org/10.1175/JAS-D-19-0257.1.

F. Lott, B. Deremble, and C. Soufflet. Mountain waves produced by a stratified shear flow with a boundary layer. part ii: Form drag, wave drag, and transition from downstream sheltering to upstream blocking.

*Journal of the Atmospheric Sciences*, 78(4):1101 – 1112, 2021. doi: https://doi.org/10.1175/JAS-D-20-0144.1.